# Measurement report: Chemical components and $^{13}$C and $^{15}$N isotope ratios of fine aerosols over Tianjin, North China: year-round observations

**Zhichao Dong, Chandra Mouli Pavuluri, Zhanjie Xu, Yu Wang, Peisen Li, Pingqing Fu**[TS1]**, and Cong-Qiang Liu**

Institute of Surface-Earth System Science, School of Earth System Science,
Tianjin University, Tianjin 300072, China

**Correspondence:** Chandra Mouli Pavuluri (cmpavuluri@tju.edu.cn)

**Abstract.** To better understand the origins and seasonality of atmospheric aerosols in North China, we collected fine aerosols (PM$_{2.5}$) at an urban site (Nankai District, ND) and a suburban site (Haihe Education Park, HEP) in Tianjin from July 2018 to July 2019. The PM$_{2.5}$ was studied for carbonaceous, nitrogenous and ionic components and stable carbon and nitrogen isotope ratios of total carbon ($\delta^{13}$C$_{TC}$) and nitrogen ($\delta^{15}$N$_{TN}$). On average, the mass concentrations of PM$_{2.5}$, organic carbon (OC), elemental carbon (EC) and water-soluble OC (WSOC) were higher in winter than in summer at both ND and HEP. SO$_4^{2-}$, NO$_3^-$ and NH$_4^+$ were the dominant ions, and their sum accounted for 89 % of the total ionic mass at ND and 87 % at HEP. NO$_3^-$ and NH$_4^+$ peaked in winter and were at their minimum in summer, whereas SO$_4^{2-}$ was higher in summer than in all the other seasons at HEP and was comparable among the seasons, although it peaked in winter at ND. $\delta^{13}$C$_{TC}$ and $\delta^{15}$N$_{TN}$ were $-26.5$‰ to $-21.9$‰ and $+1.01$‰ to $+22.8$‰ at ND and $-25.5$‰ to $-22.8$‰ and $+4.91$‰ to $+18.6$‰ at HEP. Based on seasonal variations in the measured parameters, we found that coal and biomass combustion emissions are the dominant sources of PM$_{2.5}$ in autumn and winter, while terrestrial and/or marine biological emissions are important in spring and summer in the Tianjin region, North China. In addition, our results implied that the secondary formation pathways of secondary organic aerosols in autumn/winter were different from those in spring/summer; i.e., they might be driven by NO$_3$ radicals in the former period.

## 1 Introduction

Atmospheric aerosols are mainly composed of carbonaceous and inorganic components such as elemental carbon (EC), organic matter (OM), sulfate (SO$_4^{2-}$), nitrate (NO$_3^-$), ammonium (NH$_4^+$), sea salt and minerals, each usually accounting for about 10 %–30 % of the aerosol mass load that generally ranges from 1 to 100 µg m$^{-3}$ (Pöschl, 2006; Pavuluri et al., 2015b). They have severe impacts on the Earth's climate system, air quality, visibility (Laden et al., 2000; Samet et al., 2000; Chow et al., 2002) and human health (Wessels et al., 2010). Aerosols can affect the climate directly by absorbing and scattering solar radiation and indirectly by acting as cloud condensation nuclei (CNN), and thus the hydrological cycle, at local, regional and global scales (Menon et al., 2002; Chow et al., 2006; Ramanathan et al., 2001). It has been recognized that ambient aerosol pollution is one of the major reasons for cancer (Y. Wang et al., 2016a) and other diseases in humans. According to the global burden of disease (GBD) 2010 comparative risk assessment, it has been estimated that fine aerosol (PM$_{2.5}$) pollution is causing a death rate of about 3 million people worldwide per year (Lim et al., 2012), and the total number of deaths per day is increasing by $\sim 1.5$ % for every 10 µg m$^{-3}$ increase in the average PM$_{2.5}$ loading over 2 d (Schwartz et al., 1996). Therefore, it is important to explore the source and formation processes of the PM$_{2.5}$.

Carbonaceous components, EC and organic carbon and matter (OC and OM), account for about 20 %–50 % of PM$_{2.5}$ mass (Cui et al., 2015; Sillanpää et al., 2005). EC is directly emitted from incomplete combustion of fossil fuels and biomass burning (Robinson et al., 2007; Larson and Cass, 1989), while organic aerosols (OA, generally measured as OC) can be directly emitted into the atmosphere from combustion sources, soil dust and biota (primary OC, POC) and can also be produced from volatile organic compounds (VOCs) by photochemical reactions in the atmosphere to form secondary OC (SOC) (Robinson et al., 2007). It has been estimated that OC and EC emissions have been increased by 29 % and 37 %, i.e., from 2127 and 1356 Gg in 2000 to 2749 and 1857 Gg in 2012, respectively, in China (Jimenez et al., 2009; Cui et al., 2015). Previous studies have reported very high loadings of OC and EC in large cities in China, particularly the Beijing–Tianjin–Hebei (Yang et al., 2011; Duan et al., 2005; Zhao et al., 2013; Dan et al., 2004), Yangtze River Delta (Huang et al., 2013; Feng et al., 2006, 2009; Wang et al., 2010) and Pearl River Delta (Huang et al., 2012) regions, which are densely populated and economically developed. EC has a graphite-like structure and has been recognized as a major carbonaceous component of light absorption (Zhao et al., 2013), while OC is generally considered to be a major contributor to light scattering and cooling of the atmosphere and affects cloud properties, with direct and indirect effects on the radiative forcing (Yang et al., 2011). In addition, OC contains a variety of organic compounds, such as polycyclic aromatic hydrocarbons and other harmful components that cause severe human health risks (J. Wang et al., 2016). Moreover, studies have found that the loading of SOC is significant in PM$_{2.5}$ that has been influenced by long-range atmospheric transportation of the air masses (Bikkina et al., 2017b). Many recent laboratory and field observations highlighted the importance of liquid-phase photochemical oxidation reactions in forming secondary organic aerosol (SOA) in atmospheric waters (McNeill et al., 2012; Perri et al., 2010), and hence the loading of water-soluble OC (WSOC) is increased with photochemical aging of the aerosols, which further enhances the indirect effects of the SOA.

Since industrialization, the annual production of reactive nitrogen (N$_r$) has more than doubled due to combustion of fossil fuels and production of nitrogen fertilizers and other industrial products (Gu et al., 2013). Global N$_r$ has dramatically increased from 15 Tg N yr$^{-1}$ in 1860 to 156 Tg N yr$^{-1}$ in 1995 and then to 192 Tg N yr$^{-1}$ in 2008, significantly exceeding the annual natural production from terrestrial ecosystems (40–100 Tg N yr$^{-1}$) (Gu et al., 2013). The consumption of Haber–Bosch N fixatives (HBNFs) is high (35 Tg) for agricultural and industrial applications in China, which account for about 30 % of the world's total HBNF consumption (Gu et al., 2015; Galloway et al., 2008). The N$_r$ species such as nitrogen oxides (NO$_x$: NO$_2$ and NO) and ammonia (NH$_3$) participate in a series of physical and chemical transformations, and 60 %–80 % of them convert to nitrogen-containing aerosols, affecting a variety of chemical reactions in the atmosphere (Fajardie et al., 1998). The photochemical cycle of NO$_x$ provides an important precursor for the formation of ozone. In addition, NO$_x$ can oxidize the hydrocarbons to aldehydes, ketones, acids and peroxyacetyl nitrate (PAN), leading to the formation of photochemical smog that affects the environment and human health (Wolfe and Patz, 2002). On the other hand, NH$_3$ is an important alkaline gas in the atmosphere and affects the optical properties, pH and CCN activity of aerosols and thus can influence the energy balance of the Earth's atmosphere (Bencs et al., 2010). It has also been established that secondary inorganic ions (SNA: $SO_4^{2-} + NO_3^- + NH_4^+$) are the main water-soluble inorganic ionic substances, which can directly affect the acidity of atmospheric precipitation, causing serious impacts on the ecological environment (Andreae et al., 2008) in addition to the impacts on the Earth's climate system.

Organic nitrogen (ON) is another form of N in atmospheric aerosols, such as semi-volatile amines, proteins and organic macromolecules. Water-soluble organic nitrogen (WSON), as an atmospheric input of the bioavailable nitrogen to the ecosystems, has also attracted attention in recent times (Matsumoto et al., 2018). In fact, aerosol ON is produced in the atmosphere by several secondary processes of VOCs and gaseous N species emitted from different sources (Ottley and Harrison, 1992; Utsunomiya and Wakamatsu, 1996). In addition to emissions from natural sources (such as soil and the ocean), ON can be generated by the reactions of secondary inorganic substances ($SO_4^{2-}$, $NO_3^-$, $NH_4^+$) with existing primary organic aerosol (POA) and SOA in the atmosphere. Therefore, it is difficult to understand the origins of aerosols C and N from only the measurement of their species and/or specific markers.

It is well known that the stable C ($\delta^{13}C_{TC}$) and N ($\delta^{15}N_{TN}$) isotope ratios of total C (TC) and nitrogen (TN) depend on their sources, with an obvious difference in the isotopic composition of the particles derived from different sources in the given specific area (Freyer, 1978; Moore, 1974). The particles of marine origin are highly enriched with $^{13}C$ and $^{15}N$ (Chesselet et al., 1981; Cachier et al., 1986; Miyazaki et al., 2011), which are distinct from those of the particles of continental origin, particularly anthropogenic sources such as coal combustion and vehicular emissions and the burning of C$_3$ plants as well (Cachier et al., 1986; Turekian et al., 1998; Martinelli et al., 2002; Widory, 2007; Cao et al., 2011). However, the $^{13}C$ is enriched in the particles emitted from C$_4$ plant burning, while the $^{15}N$ is enriched in those of terrestrial biogenic origin, including the biomass burning of both C$_3$ and C$_4$ plants. It has been reported that the $\delta^{13}C_{TC}$ of $-26.0$‰ and $-21.0$‰ in atmospheric aerosols represents marine and continental origins, respectively (Turekian et al., 2003; Cachier et al., 1986), while $\delta^{15}N_{TN}$ in marine aerosols ranged from $-2.2$‰ to 8.9‰ (Miyazaki et al., 2011), the particles emitted from biomass burning of different C$_3$ and

C$_4$ plants ranged from 2.0‰ to 22.7‰ and those emitted from the combustion of fossil fuels such as unleaded gasoline, diesel and coal ranged from −19.4‰ to +5.4‰ (Martinelli et al., 2002; Pavuluri et al., 2010; Widory, 2007).

On the other hand, the unidirectional chemical reactions cause an enrichment of $^{12}$C in reaction products resulting in the remaining reactants being isotopically heavier and the phase partitioning (gas to particle or vice versa, e.g., $NH_4^+ \leftrightarrow NH_3$) of a compound also resulting in isotopic fractionation (Hoefs, 1997). Furthermore, the chemical processing of aerosols results in the enrichment of $^{13}$C (and $^{15}$N) in the reaction product retained in the particle phase if some of the products are volatile (Turekian et al., 2003). Therefore, the $\delta^{13}C_{TC}$ and $\delta^{15}N_{TN}$ are modified by several chemical and physical processes in the atmosphere such as secondary aerosol formation and/or transformation (Kundu et al., 2010; Mkoma et al., 2014; Morin et al., 2009). However, such isotopic fractionation is more significant in the case of molecular species but insignificant in the case of TC and TN, because the TC and TN contents contain both the reactants and the products in the particle phase. In fact, the gas-to-particle and/or particle-to-gas transitions, even in the case of $NH_4^+ \leftrightarrow NH_3$, are not intensive, except under extreme temperatures (Pavuluri et al., 2010, 2011). Therefore, the $\delta^{13}C_{TC}$ and $\delta^{15}N_{TN}$ of PM$_{2.5}$ would provide insights, preferably into their origins. They can also provide insights into secondary formation and/or transformations of aerosols if the removal processes including physical transformation (particle to gas phase) are significant, which could result in the enrichment of $^{13}$C and $^{15}$N in the particles. Hence, the $\delta^{13}C_{TC}$ and $\delta^{15}N_{TN}$ of PM$_{2.5}$ are useful for better constraining the relative significance of such factors (Bikkina et al., 2017a; Pavuluri et al., 2010; Jickells et al., 2003; Martinelli et al., 2002). The application of $\delta^{13}$C and $\delta^{15}$N as potential tracers to investigate the origin and atmospheric processing (aging) of C and N species is well documented and has been applied in several studies in the last 2 decades (Kundu et al., 2010; Martinelli et al., 2002; Pavuluri et al., 2015c; Rudolph, 2002). However, it should be noted that the influence of isotopic fractionation by the aging on $\delta^{13}C_{TC}$ and $\delta^{15}N_{TN}$ values of PM$_{2.5}$ becomes insignificant when the local fresh air masses are mixed with the aged air masses that are transported from distant source regions and/or the aerosol removal processes are insignificant, despite the fact that the isotopic fraction must be significant at a molecular level.

Because of rapid economic growth, the aerosol loading is commonly observed to be high in China, particularly in the Beijing–Tianjin–Hebei region. According to the data analysis of the "2 + 26" list of urban industrial sources in 2018, primary emissions of PM$_{2.5}$, SO$_2$, NO$_x$ and VOCs from industrial sources account for 60 %, 46 %, 23 % and 49 % of the total regional emissions, respectively. Moreover, the total land area of Tianjin is 11 966.45 km$^2$, with the agricultural land area 6894.41 km$^2$, accounting for 57.6 % of the total land area. According to the results of the 9th China forest resources inventory, Tianjin has 2039 km$^2$ of forest area (17.0 % of the total land area). In addition, there are 17 natural protected areas of various types, with a total area of about 1418.79 km$^2$ in Tianjin (https://www.tjrd.gov.cn/tjfq/system/2019/04/24/030012397.shtml, last access: 24 April 2019 TS2). Thus, Tianjin is surrounded by areas largely covered with agricultural fields and forests that emit large amounts of VOCs and bioaerosols. On the other hand, the East Asian monsoon climate prevailing over the region brings the long-range transported air masses to Tianjin, and their origins vary with the season (Wang et al., 2018). Therefore, the investigation of the Tianjin aerosols sources and formation processes provides better insights into the types of aerosol sources at a regional level, in addition to the local industrial and domestic pollutant emissions in North China. However, the studies on Tianjin aerosols are limited and mostly focused on the short-term measurements of mass concentrations of PM$_{2.5}$, EC, OC and/or inorganic ions (Kong et al., 2010; Li et al., 2009, 2012; X. Li et al., 2017) but not the long-term measurements and seasonal characterization of carbonaceous and nitrogenous components and water-soluble inorganic ions that are important for better understanding the sources and characteristics of the PM$_{2.5}$ (Cao et al., 2007; Dentener et al., 2006; Pavuluri et al., 2015b). Furthermore, ON, which represents a significant fraction (up to 80 %) of total aerosol N and plays an important role in biogeochemical cycles (Pavuluri et al., 2015a; Cape et al., 2011), has not been studied in Tianjin aerosols.

Therefore, the comprehensive study of various chemical components and $\delta^{13}C_{TC}$ and $\delta^{15}N_{TN}$ of PM$_{2.5}$ in Tianjin is highly needed in order to better understand their origins and even aging to some extent over the region. Here, we present the characteristics and seasonality of carbonaceous (EC, OC, WSOC, WIOC and SOC) and nitrogenous (IN, ON and WSON) components, inorganic ions (Cl$^-$, SO$_4^{2-}$, NO$_3^-$, Na$^+$, NH$_4^+$, K$^+$, Mg$^{2+}$ and Ca$^{2+}$) and $\delta^{13}C_{TC}$ and $\delta^{15}N_{TN}$ in PM$_{2.5}$ collected over a 1-year period at an urban site and a suburban site in Tianjin, North China. Based on the chemical compositions, $\delta^{15}N_{TN}$ and $\delta^{13}C_{TC}$ and their seasonal changes, we discuss the origins and possible aging of PM$_{2.5}$ over the Tianjin region.

## 2 Materials and methods

### 2.1 Aerosol sampling and mass measurement

PM$_{2.5}$ sampling was performed at an urban site, Nankai District (ND), located in the central part at 39.11° N, 117.18° E, and a suburban (background) site, Haihe Education Park (HEP), located at 39.00° N, 117.32° E, 23 km away from ND, in Tianjin, a coastal metropolis located on the lower reaches of the Haihe River and the Bohai Sea in the Beijing–Tianjin–Hebei urban economic area in the north-

ern part of the Chinese mainland (Fig. 1), with a population of ∼ 16 million (https://wiki.hk.wjbk.site, last access: 24 April 2019). The PM$_{2.5}$ sampling was conducted on the rooftop of a seven-storey teaching building of Tianjin University Weijin Road campus in ND for about 72 h (3 consecutive days), each sampled continuously from 5 July 2018 to 4 July 2019 using precombusted (450 °C, 6 h) quartz membrane (Pallflex 2500QAT-UP) filters and a high-volume air sampler (Tisch Environmental, TE-6070DX) with a flow rate of $1.0\,m^3\,min^{-1}$ ($n = 121$). Simultaneously, the PM$_{2.5}$ sampling was conducted on the rooftop of a six-storey teaching building of Tianjin University Peiyangyuan campus in HEP with the same sampling frequency (72 h each) for a 1-month period in each season: from 5 July–4 August in summer, 30 September–30 October in autumn 2018, 1 January–1 February in winter and 2 April–2 May in spring 2019. Prior to analysis, the filter samples were placed in a precombusted glass jar with a Teflon-lined cap and stored in the dark at −20 °C. A blank filter sample was also collected in each season following the same procedure without turning on the sampler pump and placing the filter in a filter hood for 10 min.

Each filter was dehumidified in a desiccator for 48 h before and after sampling, and the mass concentration of PM$_{2.5}$ was determined by gravimetric analysis.

It should be noted that PM$_{2.5}$ samples collected on quartz fiber filters might have positive sampling artifacts due to the adsorption of gas-phase organic and nitrogen compounds and the negative artifacts by evaporation of the semi-volatile organic and nitrogen species from the aerosol particles (Turpin et al., 2000; Schaap et al., 2004). Since the sampling time is long (∼ 72 h) in this study, the evaporation of semi-volatile species from the particles should be more effective than the adsorption of gaseous species by a quartz fiber filter, which would be saturated upon continuous sampling, and thus the reported concentrations may be underestimated. However, we consider that such losses should be minimal because the ambient temperatures encountered in Tianjin are rather low (see Sect. 3.1) and thus may not cause a significant evaporative loss (Schaap et al., 2004) during the sampling period. Therefore, we believe that our sampling technique does not have serious sampling artifacts even in summer, although we do not rule them out completely.

## 2.2 Chemical analyses

### 2.2.1 Measurements of carbonaceous components

OC and EC were measured using an OC/EC analyzer (USA, Sunset Laboratory Inc.) based on thermal light transmission following the IMPROVE protocol of the protective visual environment (Wan et al., 2017, 2015; Chow et al., 2007) and assuming the carbonate carbon was negligible (Pavuluri et al., 2011; Wang et al., 2019), because the C removed by HCl treatment has been reported to be only 6.3 % in TC at Gosan

Island, South Korea (Kawamura et al., 2004), where the long-range transported air masses enriched with soil dust are the major sources, rather than anthropogenic sources, unlike in the Tianjin region. Briefly, an aliquot of a filter (1.5 cm$^2$) of each sample was punched and placed in a quartz boat in the thermal desorption chamber of the analyzer, and then the carbon content of each sample was measured by a two-step heating procedure. The analytical principle of the instrument has been described in detail in the literature (Cao et al., 2007; Watson et al., 2005). During the experiment, a sucrose solution with a known carbon content (36.1 ± 1.8 μg C TS3) was used as a standard reference for the measurement of OC and EC. The analytical errors in duplicate analyses were within 2 % for OC and 5 % for EC.

The total organic carbon (TOC) analyzer (model: OI, 1030W + 1088) was used to measure the content of WSOC. Total inorganic carbon (TIC) obtained by acidizing the sample with HCl down to a pH of less than 2.0 and TOC obtained by wet oxidation, i.e., oxidizing the sample with an agent (e.g., persulfate) at 100 °C, can be measured simultaneously with the same sample, ensuring the highest detection accuracy and reliability of the data. An aliquot of a filter sample (one disc of 14 mm in diameter for filters 1–65 and 22 mm for filters 66–172) was extracted into 20 and 30 mL organic-free Milli-Q water, respectively, under ultrasonication for 20 min (Wang et al., 2019). The extracts were filtered through a 0.22 μm polytetrafluoroethylene (PTFE) syringe filter, and then the content of the WSOC was measured using the TOC analyzer. The analytical uncertainty in measurements was generally less than 5 %. The concentrations of OC, EC and WSOC were corrected for field blanks.

The sum of OC and EC was considered to be TC, and the difference between OC and WSOC was considered to be the water-insoluble OC (WIOC) (Wang et al., 2018).

Due to a lack of analytical methods to directly measure SOC (Turpin and Huntzicker, 1995), the SOC was estimated using the OC/EC tracer-based method proposed by Turpin et al. (2000). The formula for its calculation is as follows:

$$SOC = OC - (EC \times (OC/EC)_{pri}),$$

where $(OC/EC)_{pri}$ is the mass concentration ratio between OC and EC generated by primary emission, which is generally the minimum value among the measured OC/EC. Because the OC/EC is highly influenced by meteorological conditions, emission sources and other factors and thus the estimation of SOC using the minimum value results in a large deviation, we used the average value of three minimum values in the OC/EC ratios as the $(OC/EC)_{pri}$, which was 6.71 at ND and 4.62 at HEP.

### 2.2.2 Measurements of inorganic ions

Inorganic ions Cl$^-$, SO$_4^{2-}$, NO$_3^-$, Na$^+$, NH$_4^+$, K$^+$, Mg$^{2+}$ and Ca$^{2+}$ were measured using ion chromatography (ICS-5000 System, China, Dai An). An aliquot of a filter sample

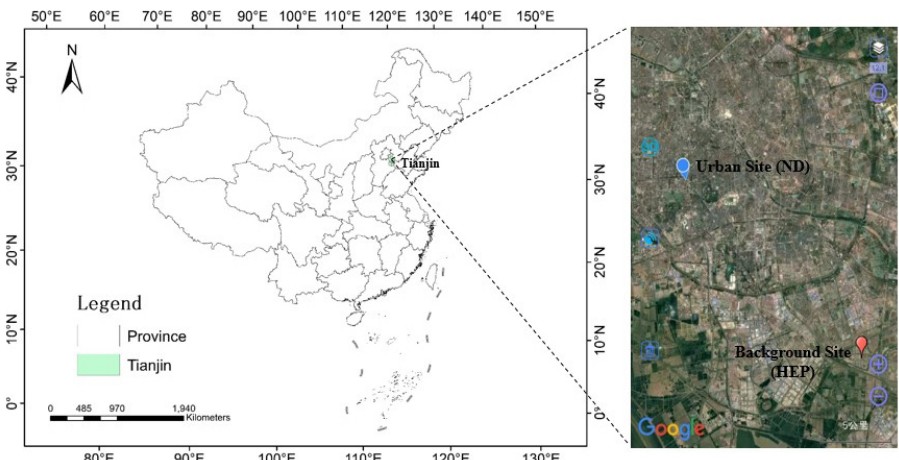

**Figure 1.** Map of China with the study area, Tianjin, North China. The sampling points located in urban (ND) and suburban (HEP) locations are shown in the inset. (The map in the inset was generated using © Google Maps.)

(one disc of 22 mm in diameter) was extracted into 30 mL Milli-Q water under ultrasonication for 30 min and filtered with a PTFE syringe filter (0.22 μm) and then injected into an ion chromatograph. A mixture of $NaHCO_3$, $Na_2CO_3$, NaOH (50 % NaOH solution) eluent and an IonPac AG11-HC/AS11-HC column and a 30 mM KOH suppresser with a flow rate of 1.2 mL min$^{-1}$ was used for anion measurement. For cationic measurement, methyl sulfonic acid and IonPac CS12A and CG12A columns at a flow rate of 1.0 mL min$^{-1}$ with a 20 mM mesylate suppresser were used. The analytical error in duplicate analyses was generally less than 5 %. Concentrations of all the ions were corrected for field blanks.

### 2.2.3 Determination of nitrogenous components

Water-soluble total nitrogen (WSTN) was determined using a continuous-flow analyzer (CFA, Skalar, the Netherlands, San++) following the standard procedure. Briefly, an aliquot of a filter (3.80 cm$^2$) sample was extracted into 10 mL Milli-Q water under ultrasonication for 10 min each three times, and the extracts were filtered through 0.22 μm-sized Teflon syringe filters to remove filter debris. The filter extracts were then mixed with excess potassium persulfate and digested in the UV digester to convert all N to $NO_3^-$ and passed through a reduction column equipped with a granular copper and cadmium column to reduce $NO_3^-$ to nitrite ($NO_2^-$). The produced nitrite is reacted with aminobenzene sulfonic acid to result in high molecular weight nitrogen compounds (azo dye), and then the absorbance of total N was measured at 540 nm. The average analysis error of the repeated analysis was 167.1 %. Such a large analytical error can be attributed to the slightly low reproducibility of the instrument with the detection limit of 0.01–5 mg L$^{-1}$ and the uneven distribution of particles in the sampling filter.

The N contents of $NO_2^-$, $NO_3^-$ and $NH_4^+$ were calculated from their concentrations by multiplying them by the percent factor of N in the given molecule. The sum of those contents was considered to be the total inorganic nitrogen (IN). The difference between the concentrations of WSTN and IN was considered to be WSON (Matsumoto et al., 2018):

$$WSON = WSTN - IN.$$

However, it should be noted that the analytical uncertainties associated with the measurements of WSTN and $NO_2^-$, $NO_3^-$ and $NH_4^+$ must result in huge error in the estimation of WSON. The propagating error in WSON estimation from duplicate analysis of the samples for $NO_3^-$, $NH_4^+$ and WSTN with uncertainties of 0.78 %, 1.82 % and 16.1 %, respectively, was 0.83. However, we consider that such errors may not influence the conclusions drawn from this study, because they were drawn based on the temporal trends rather than the absolute concentrations.

### 2.2.4 Determination of stable carbon and nitrogen isotope ratios of TC ($\delta^{13}C_{TC}$) and TN ($\delta^{15}N_{TN}$)

$\delta^{13}C_{TC}$ and $\delta^{15}N_{TN}$ were determined using an elemental analyzer (EA, Flash 2000HT) coupled with a stable isotope ratio mass spectrometer (IRMS, 253 Plus). Briefly, an aliquot of a filter was subjected to acid steaming and placed in a dry dish containing concentrated $HNO_3$ for 12 h to remove inorganic carbon ($CaCO_3$) content, if any, which affects the result of the $\delta^{13}C_{TC}$. The acidified filter sample was dried out in an oven for 24 h and then packed into a tin cup that was injected into EA. The derived gases $CO_2$ and $N_2$ were transferred into IRMS through ConFlo-II to measure the $^{13}C/^{12}C$ in TC and $^{15}N/^{14}N$ in TN.

The isotope ratios of $^{13}C/^{12}C$ and $^{15}N/^{14}N$ are expressed as delta ($\delta$) values ($\delta^{13}C_{TC}$ and $\delta^{15}N_{TN}$) after the normalization with Pee Dee Belemnite (PDB) and atmospheric $N_2$ in parts per million (Duarte et al., 2019; Li et al., 2022). The

$\delta^{13}C_{TC}$ and $\delta^{15}N_{TN}$ were calculated using the following for-
mulas (Fu et al., 2012; Pavuluri et al., 2010):

$$\delta^{13}C_{TC} = [(^{13}C/^{12}C)_{sample}/(^{13}C/^{12}C)_{standard} - 1] \times 1000,$$
$$\delta^{15}N_{TN} = [(^{15}N/^{14}N)_{sample}/(^{15}N/^{14}N)_{standard} - 1] \times 1000.$$

### 2.3 Measurements of meteorological parameters, simulations of air mass trajectories, and data statistical analyses

The meteorology data from Tianjin were collected from a mobile weather station (Gill MetPak, UK) installed at the sampling site during the campaign in this study. Five-day backward air mass trajectory clustering analysis was conducted using the NOAA HYLSPLIT modeling system (https://www.ready.noaa.gov/HYSPLIT.php, last access: 15 June 2020) for every month to identify the source regions of the air parcels that arrived over Tianjin at 500 m a.g.l. (above the ground level) at a regional scale during the campaign. The statistical analysis of the obtained data was performed using Igor Pro 7 (WaveMatrics Inc., OR, USA) software. A summary of the statistics and linear analysis has been carried out in order to characterize the variations in individual measurements as well as to estimate the relation between the considered parameters.

## 3 Results and discussion

### 3.1 Meteorology and backward air mass trajectories

Temporal variations in the averages of the data for each sample period are depicted in Fig. 2. The ambient temperature, relative humidity (RH) and wind speed showed a clear seasonal pattern (Fig. 2). On average, the temperature was higher (27.3 °C) in summer and lower (1.28 °C) in winter. The annual average of RH was 39.2 %. It was relatively higher in summer and autumn than that in winter and spring. The average wind speed in autumn (2.03 m s$^{-1}$) was almost similar to that in spring (2.06 m s$^{-1}$) but was lower in summer (1.64 m s$^{-1}$) and winter (1.58 m s$^{-1}$).

Plots of 5 d backward air mass trajectory clusters are depicted in Fig. 3. The trajectories showed that most of the air masses that arrived in Tianjin originated from the oceanic region in summer (Fig. 3). In particular, 50 % of the air masses originated from the East Sea in July 2018, while a small portion (8 %) of the air masses originated from Northeast China and Siberia, whereas in autumn, winter and spring, they mostly originated from Siberia and Mongolia as well as from inland China (Fig. 3). It is noteworthy that 33 % of the air masses arrived in Tianjin during September, and 28 % during October originated from the northern parts of China. Therefore, the chemical composition and characteristics of PM$_{2.5}$ in Tianjin should have been significantly influenced by the long-range transported air masses and varied according to seasonal changes in addition to the local emissions.

### 3.2 Concentrations and seasonal variations of PM$_{2.5}$

Concentrations of PM$_{2.5}$ and its carbonaceous components EC, OC, SOC, WSOC, WIOC and TC, nitrogenous components WSTN, IN and WSON, inorganic ions (Cl$^-$, NO$_3^-$, SO$_4^{2-}$, Na$^+$, K$^+$, NH$_4^+$, Ca$^{2+}$ and Mg$^{2+}$) as well as $\delta^{13}C_{TC}$ and $\delta^{15}C_{TN}$ during the whole campaign (annual) and in each season at ND and HEP in Tianjin, North China, are summarized in Table 1. Generally, PM$_{2.5}$ levels are controlled by emissions, transport and chemical transformation and deposition processes, all of which are influenced by meteorological conditions (Yang et al., 2011; Zhang et al., 2013). The temporal trend of PM$_{2.5}$ was found to be similar to that of RH and opposite to that of wind speed (Figs. 2 and 4). On average, the concentrations of PM$_{2.5}$ at ND and HEP in winter were 4 and 3 times higher than that in summer (Table 1). According to the China ambient air quality standard (GB3095-2012), the average PM$_{2.5}$ concentration limit in the ambient environment is 75 μg m$^{-3}$ for 24 h and 35 μg m$^{-3}$ for the year. Although the annual average concentration of PM$_{2.5}$ in Tianjin did not exceed the national PM$_{2.5}$ limit, it is about 3 times higher than that of the global limit (10 μg m$^{-3}$) stipulated by the World Health Organization. Furthermore, the average concentration of PM$_{2.5}$ was found to be higher in spring than in autumn (Table 1), probably due to enhanced eruption of dust from open lands by relatively strong winds in spring (Fig. 2). In addition, the long-range transported air masses that passed over the Mongolian region must have been enriched with the soil dust, resulting in the higher levels of PM$_{2.5}$ in spring in Tianjin. In fact, the dust storms over Mongolia and China that are common in spring enhance the loading of PM$_{2.5}$ in the East Asian atmosphere (Liu et al., 2011).

However, the average PM$_{2.5}$ concentration found in this study is significantly lower than that (109.8 μg m$^{-3}$) reported 10 years before in Tianjin (Li et al., 2009). Furthermore, it is also lower than that reported in Harbin, Northeast China, but similar to that recorded in the southeastern coastal cities in China: Ningbo and Guangzhou (Table 2). Such a relatively lower concentration of PM$_{2.5}$ in Tianjin is likely, because the government is strictly implementing various control measures, including the replacement of coal with natural gas and electricity (http://huanbao.bjx.com.cn/news/20170901/847140.shtml, last access: 1 September 2017) in air pollutant emissions in North China since 2013. It has been reported that the average concentration of PM$_{2.5}$ decreased from 2011 to 2017 in the southwestern city of Chengdu, consistent with the variation trend of PM$_{2.5}$ concentrations in most cities in North China (Table 2), which indicates that the government's measures for prevention and control of air pollution are effective in China. However, the PM$_{2.5}$ loading over most Chinese cities, including Tianjin, is still much higher compared to that reported in the USA (Table 2). Such comparisons indicate that the aerosol loading is significantly high in the Tianjin atmosphere and needs to continue the en-

**Table 1.** Summary of concentrations of carbonaceous (EC, OC, SOC, WSOC, WIOC and TC), nitrogenous (WSTN, IN and WSON) and inorganic ionic (Cl$^-$, NO$_3^-$, SO$_4^{2-}$, Na$^+$, K$^+$, NH$_4^+$, Ca$^{2+}$ and Mg$^{2+}$) components (µg m$^{-3}$) and stable carbon and nitrogen isotope ratios (‰) of total carbon ($\delta^{13}$C$_{TC}$) and nitrogen ($\delta^{15}$C$_{TN}$) in fine aerosols together with the PM$_{2.5}$ mass (µg m$^{-3}$) at urban (ND) and suburban (HEP) sites in Tianjin, North China, on 5 July 2018 and 5 July 2019.

| Components | Annual | | Summer | | Autumn | | Winter | | Spring | |
|---|---|---|---|---|---|---|---|---|---|---|
| | Range/med | Avg ± SD | Range/med | Avg ± SD | Range/med | Avg ± SD | Range/med | Avg ± SD | Range/med | Avg ± SD |
| | ND (n = 121)<br>HEP (n = 40) | | ND (n = 30, Jun–Aug)<br>HEP (n = 10, Jul) | | ND (n = 30, Sep–Nov)<br>HEP (n = 10, Oct) | | ND (n = 30, Dec–Feb)<br>HEP (n = 10, Jan) | | ND (n = 31, Mar–May)<br>HEP (n = 10, Apr) | |
| Carbonaceous components (µg m$^{-3}$) | | | | | | | | | | |
| EC | 0.10 to 0.56/0.26<br>0.09 to 0.81/0.40 | 0.27 ± 0.11<br>0.40 ± 0.18 | 0.11 to 0.31/0.18<br>0.09 to 0.59/0.27 | 0.18 ± 0.05<br>0.28 ± 0.16 | 0.21 to 0.54/0.33<br>0.41 to 0.81/0.61 | 0.36 ± 0.10<br>0.59 ± 0.13 | 0.10 to 0.56/0.28<br>0.15 to 0.53/0.39 | 0.30 ± 0.10<br>0.37 ± 0.10 | 0.10 to 0.34/0.25<br>0.18 to −0.62/0.32 | 0.24 ± 0.07<br>0.36 ± 0.15 |
| OC | 1.37 to 24.7/3.40<br>0.85 to 14.7/4.40 | 4.93 ± 3.79<br>5.61 ± 3.55 | 1.37 to 3.26/2.31<br>0.85 to 4.34/2.25 | 2.31 ± 0.52<br>2.44 ± 1.20 | 1.48 to 12.8/4.44<br>3.01 to 9.86/4.62 | 5.00 ± 2.65<br>5.28 ± 2.07 | 2.49 to 24.7/7.97<br>7.18 to 14.7/9.51 | 8.79 ± 4.85<br>10.4 ± 2.98 | 1.52 to 6.58/3.38<br>2.46 to 5.68/4.39 | 3.36 ± 1.12<br>4.30 ± 1.11 |
| WSOC | 0.69 to 16.0/2.56<br>0.66 to 9.44/3.52 | 3.25 ± 2.18<br>3.47 ± 2.04 | 1.14 to 3.12/1.74<br>0.66 to 3.73/1.81 | 1.88 ± 0.53<br>2.16 ± 1.17 | 1.16 to 7.68/3.13<br>1.48 to 6.11/2.93 | 3.45 ± 1.74<br>3.08 ± 1.41 | 1.37 to 16.0/4.19<br>4.00 to 9.44/5.50 | 5.06 ± 2.99<br>4.00 ± 9.44 | 0.69 to 4.03/2.44<br>0.95 to 4.38/2.44 | 2.48 ± 0.82<br>2.70 ± 1.18 |
| WIOC | 0.00 to 8.93/1.01<br>0.00 to 7.39/1.77 | 1.68 ± 1.77<br>2.14 ± 1.75 | 0.00 to 1.33/0.38<br>0.00 to 0.67/0.22 | 0.43 ± 0.32<br>0.29 ± 0.21 | 0.21 to 5.07/1.37<br>0.61 to 3.75/2.26 | 1.55 ± 1.04<br>2.20 ± 0.86 | 0.00 to 8.93/3.33<br>3.02 to 7.39/3.93 | 3.74 ± 2.09<br>4.48 ± 1.45 | 0.23 to 2.62/0.73<br>0.51 to 3.15/1.43 | 0.88 ± 0.63<br>1.60 ± 0.70 |
| SOC | 0.00 to 20.9/1.65<br>0.00 to 12.8/2.86 | 3.11 ± 3.43<br>3.75 ± 3.48 | 0.24 to 1.75/1.13<br>0.00 to 2.97/0.71 | 1.08 ± 0.37<br>1.18 ± 1.01 | 0.00 to 10.7/2.24<br>0.05 to 6.73/2.56 | 2.59 ± 2.58<br>2.56 ± 2.15 | 1.58 to 20.9/5.92<br>5.46 to 12.8/8.48 | 6.78 ± 4.35<br>8.68 ± 2.90 | 0.83 to 4.88/1.51<br>1.04 to 4.18/2.67 | 1.73 ± 0.87<br>2.63 ± 1.08 |
| WSOC/OC | 0.30 to 1.05/0.72<br>0.30 to 1.04/0.61 | 0.71 ± 0.15<br>0.66 ± 0.17 | 0.54 to 1.05/0.80<br>0.77 to 1.04/0.84 | 0.82 ± 0.12<br>0.87 ± 0.09 | 0.43 to 0.92/0.70<br>0.47 to 0.85/0.57 | 0.70 ± 0.10<br>0.58 ± 0.11 | 0.41 to 1.05/0.57<br>0.46 to 0.67/0.58 | 0.57 ± 0.11<br>0.57 ± 0.07 | 0.30 to 0.93/0.79<br>0.30 to 0.82/0.62 | 0.75 ± 0.14<br>0.61 ± 0.17 |
| OC/EC | 6.56 to 48.1/14.4<br>4.01 to 63.0/12.4 | 17.8 ± 9.46<br>15.7 ± 11.5 | 7.64 to 16.2/13.3<br>4.01 to 14.9/10.1 | 13.1 ± 2.18<br>9.67 ± 3.62 | 6.56 to 41.7/12.5<br>4.67 to 14.5/10.3 | 14.5 ± 8.39<br>9.31 ± 3.70 | 16.1 to 48.1/28.1<br>16.1 to 63.0/29.5 | 29.2 ± 9.73<br>30.4 ± 13.1 | 9.37 to 25.9/13.3<br>6.28 to 19.7/13.2 | 14.1 ± 3.18<br>13.2 ± 4.70 |
| SOC/OC | 0.00 to 0.86/0.53<br>0.00 to 0.93/0.63 | 0.53 ± 0.20<br>0.57 ± 0.25 | 0.12 to 0.58/0.49<br>0.00 to 0.69/0.54 | 0.47 ± 0.11<br>0.46 ± 0.22 | 0.00 to 0.84/0.46<br>0.01 to 0.68/0.55 | 0.41 ± 0.26<br>0.41 ± 0.26 | 0.58 to 0.86/0.76<br>0.71 to 0.93/0.84 | 0.74 ± 0.08<br>0.82 ± 0.07 | 0.28 to 0.74/0.49<br>0.27 to 0.77/0.65 | 0.50 ± 0.09<br>0.60 ± 0.16 |
| Nitrogenous components (µg m$^{-3}$) | | | | | | | | | | |
| WSTN | 0.32 to 26.3/3.15<br>1.34 to 18.4/6.25 | 5.45 ± 5.50<br>7.34 ± 5.13 | 0.56 to 4.57/1.73<br>1.37 to 6.61/3.93 | 1.77 ± 0.86<br>3.64 ± 1.57 | 0.32 to 24.9/5.47<br>1.57 to 18.2/5.92 | 6.63 ± 6.06<br>7.23 ± 5.90 | 1.52 to 26.3/5.84<br>4.68 to 18.4/7.94 | 8.51 ± 6.40<br>9.68 ± 4.20 | 0.73 to 16.1/3.33<br>1.34 to 17.5/8.57 | 4.80 ± 4.03<br>8.80 ± 5.67 |
| IN | 0.00 to 26.5/3.35<br>1.39 to 14.8/5.40 | 5.21 ± 5.01<br>6.14 ± 3.90 | 0.00 to 5.67/1.79<br>1.40 to 5.43/3.54 | 1.82 ± 0.96<br>3.32 ± 1.17 | 0.19 to 21.3/5.32<br>1.39 to 14.5/6.43 | 6.10 ± 5.38<br>6.45 ± 4.61 | 1.86 to 26.5/5.82<br>4.24 to 14.2/6.81 | 8.23 ± 5.91<br>7.98 ± 3.08 | 1.00 to 16.0/3.50<br>1.61 to 14.8/5.97 | 4.68 ± 3.55<br>6.82 ± 4.32 |
| WSON | 0.00 to 3.51/0.72<br>0.00 to 9.80/0.77 | 0.40 ± 0.69<br>1.29 ± 1.47 | 0.00 to 0.39/0.03<br>0.00 to 1.18/0.43 | 0.07 ± 0.09<br>0.47 ± 0.36 | 0.00 to 3.51/0.31<br>0.00 to 3.65/0.10 | 0.63 ± 0.83<br>1.01 ± 1.46 | 0.00 to 2.32/0.01<br>0.44 to 4.18/1.18 | 0.40 ± 0.65<br>1.70 ± 1.30 | 0.00 to 3.14/0.17<br>0.00 to 6.03/1.57 | 0.50 ± 0.77<br>2.01 ± 1.90 |
| WSON/WETN | 0.00 to 0.40/0.07<br>0.00 to 0.48/0.16 | 0.07 ± 0.08<br>0.14 ± 0.10 | 0.00 to 0.17/0.02<br>0.00 to 0.18/0.12 | 0.05 ± 0.06<br>0.11 ± 0.07 | 0.00 to 0.40/0.10<br>0.00 to 0.22/0.05 | 0.12 ± 0.10<br>0.09 ± 0.10 | 0.00 to 0.17/0.00<br>0.09 to 0.31/0.14 | 0.03 ± 0.04<br>0.16 ± 0.06 | 0.00 to 0.31/0.06<br>0.00 to 0.48/0.16 | 0.07 ± 0.08<br>0.19 ± 0.12 |

| Components | Annual Range/med | Annual Avg ± SD | Summer Range/med | Summer Avg ± SD | Autumn Range/med | Autumn Avg ± SD | Winter Range/med | Winter Avg ± SD | Spring Range/med | Spring Avg ± SD |
|---|---|---|---|---|---|---|---|---|---|---|
| | ND ($n = 121$)<br>HEP ($n = 40$) | | ND ($n = 30$, Jun–Aug)<br>HEP ($n = 10$, Jul) | | ND ($n = 30$, Sep–Nov)<br>HEP ($n = 10$, Oct) | | ND ($n = 30$, Dec–Feb)<br>HEP ($n = 10$, Jan) | | ND ($n = 31$, Mar–May)<br>HEP ($n = 10$, Apr) | |
| **Inorganic ions (µg m$^{-3}$)** | | | | | | | | | | |
| Cl$^-$ | 0.01 to 9.22/0.68<br>0.04 to 6.83/1.25 | 1.44 ± 1.80<br>1.87 ± 1.91 | 0.02 to 0.130/0.06<br>0.04 to 0.370/0.08 | 0.07 ± 0.03<br>0.14 ± 0.12 | 0.01 to 4.97/1.36<br>0.11 to 4.05/1.84 | 1.46 ± 1.45<br>1.90 ± 1.07 | 0.64 to 9.22/3.30<br>2.70 to 6.83/4.08 | 3.49 ± 1.94<br>4.56 ± 1.34 | 0.07 to 2.22/0.57<br>0.20 to 2.29/0.80 | 0.64 ± 0.55<br>0.87 ± 0.62 |
| SO$_4^{2-}$ | 0.50 to 21.6/3.73<br>1.00 to 15.0/5.44 | 4.56 ± 3.32<br>5.93 ± 3.78 | 1.79 to 8.81/4.43<br>3.09 to 15.0/9.18 | 4.42 ± 1.73<br>9.21 ± 4.63 | 0.50 to 12.8/4.58<br>1.00 to 8.57/5.44 | 4.39 ± 2.91<br>4.30 ± 2.68 | 1.21 to 21.6/3.26<br>2.29 to 12.0/3.79 | 5.62 ± 5.05<br>4.93 ± 3.00 | 0.99 to 9.15/3.13<br>2.00 to 10.9/5.03 | 3.55 ± 2.00<br>5.28 ± 2.77 |
| NO$_3^-$ | 0.08 to 37.7/4.69<br>0.18 to 27.6/6.35 | 7.38 ± 8.16<br>8.59 ± 7.57 | 0.08 to 8.85/0.33<br>0.18 to 5.59/1.21 | 0.91 ± 1.65<br>2.06 ± 1.98 | 0.13 to 31.8/8.11<br>1.35 to 27.6/11.0 | 9.90 ± 9.41<br>11.4 ± 9.63 | 2.26 to 37.7/8.38<br>4.68 to 18.6/9.82 | 11.1 ± 8.29<br>10.7 ± 4.66 | 0.74 to 21.0/4.91<br>1.91 to 24.5/7.55 | 6.90 ± 5.85<br>10.2 ± 8.14 |
| Na$^+$ | 0.00 to 0.80/0.11<br>0.01 to 0.37/0.15 | 0.15 ± 0.14<br>0.16 ± 0.09 | 0.00 to 0.270/0.06<br>0.11 to 0.220/0.14 | 0.09 ± 0.06<br>0.16 ± 0.04 | 0.01 to 0.38/0.20<br>0.15 to 0.33/0.23 | 0.19 ± 0.11<br>0.23 ± 0.06 | 0.00 to 0.80/0.22<br>0.02 to 0.37/0.12 | 0.27 ± 0.20<br>0.15 ± 0.11 | 0.00 to 0.24/0.07<br>0.01 to 0.25/0.10 | 0.08 ± 0.07<br>0.11 ± 0.08 |
| Mg$^{2+}$ | 0.00 to 0.36/0.03<br>0.00 to 0.15/0.03 | 0.03 ± 0.04<br>0.03 ± 0.04 | 0.00 to 0.060/0.00<br>0.00 to 0.030/0.03 | 0.00 ± 0.01<br>0.00 ± 0.01 | 0.00 to 0.06/0.00<br>0.00 to 0.00/0.00 | 0.01 ± 0.02<br>0.00 ± 0.00 | 0.02 to 0.36/0.04<br>0.04 to 0.15/0.06 | 0.06 ± 0.07<br>0.07 ± 0.03 | 0.00 to 0.06/0.03<br>0.00 to 0.11/0.03 | 0.03 ± 0.02<br>0.04 ± 0.04 |
| K$^+$ | 0.03 to 3.83/0.29<br>0.09 to 1.27/0.39 | 0.48 ± 0.53<br>0.49 ± 0.31 | 0.06 to 0.230/0.12<br>0.09 to 0.330/0.24 | 0.13 ± 0.05<br>0.21 ± 0.09 | 0.03 to 1.17/0.45<br>0.24 to 1.05/0.54 | 0.49 ± 0.36<br>0.55 ± 0.28 | 0.16 to 3.83/0.67<br>0.56 to 1.27/0.79 | 0.96 ± 0.77<br>0.84 ± 0.24 | 0.07 to 0.56/0.27<br>0.15 to 0.68/0.32 | 0.29 ± 0.14<br>0.38 ± 0.19 |
| NH$_4^+$ | 0.19 to 23.2/3.01<br>1.06 to 13.1/5.02 | 4.59 ± 4.12<br>5.40 ± 3.08 | 0.62 to 4.72/2.17<br>1.42 to 5.35/3.93 | 2.08 ± 0.90<br>3.67 ± 1.43 | 0.19 to 18.2/4.48<br>1.06 to 10.6/5.07 | 4.97 ± 4.35<br>4.99 ± 3.46 | 1.73 to 23.2/5.26<br>4.09 to 13.1/5.93 | 6.92 ± 5.05<br>7.16 ± 2.88 | 0.97 to 14.5/3.10<br>1.52 to 11.9/5.48 | 4.01 ± 2.92<br>5.79 ± 3.41 |
| Ca$^{2+}$ | 0.00 to 0.81/0.13<br>0.00 to 1.08/0.17 | 0.11 ± 0.12<br>0.22 ± 0.23 | 0.00 to 0.300/0.00<br>0.00 to 0.41/0.05 | 0.02 ± 0.06<br>0.08 ± 0.12 | 0.00 to 0.32/0.01<br>0.14 to 0.21/0.10 | 0.07 ± 0.08<br>0.11 ± 0.05 | 0.05 to 0.81/0.15<br>0.14 to 1.08/0.30 | 0.20 ± 0.14<br>0.37 ± 0.26 | 0.02 to 0.47/0.11<br>0.01 to 0.85/0.29 | 0.14 ± 0.11<br>0.34 ± 0.25 |
| **Isotope ratios (‰)** | | | | | | | | | | |
| δ$^{15}$N$_{TN}$ | 1.01 to 22.8/10.2<br>4.91 to 18.6/9.75 | 11.4 ± 4.83<br>10.4 ± 3.43 | 13.2 to 22.8/17.4<br>6.86 to 18.6/14.8 | 17.6 ± 2.52<br>14.5 ± 3.46 | 2.93 to 20.2/9.90<br>5.61 to 12.6/9.49 | 10.4 ± 4.52<br>8.78 ± 2.27 | 1.01 to 11.8/8.86<br>4.91 to 11.9/8.66 | 8.21 ± 2.49<br>8.41 ± 2.12 | 5.06 to 16.1/9.79<br>7.01 to 12.2/10.0 | 9.82 ± 2.72<br>9.94 ± 1.66 |
| δ$^{13}$C$_{TC}$ | −26.5 to<br>−21.9/−25.2<br>−25.5 to<br>−22.8/−24.5 | −25.0 ± 0.70<br>−24.5 ± 0.55 | −26.0 to<br>−25.1/−25.6<br>−25.1 to<br>−24.1/−24.8 | −25.6 ± 0.26<br>−24.7 ± 0.30 | −25.7 to<br>−21.9/−24.9<br>−25.1 to<br>−22.8/−24.0 | −24.7 ± 0.81<br>−23.9 ± 0.60 | −25.9 to<br>−23.7/−24.5<br>−25.1 to<br>−24.1/−24.5 | −24.5 ± 0.48<br>−24.5 ± 0.29 | −26.5 to<br>−24.4/−25.4<br>−25.5 to<br>−24.3/−24.9 | −25.4 ± 0.53<br>−24.9 ± 0.34 |
| **Aerosol mass (µg m$^{-3}$)** | | | | | | | | | | |
| PM$_{2.5}$ | 3.38 to 170/23.6<br>7.56 to 103/38.9 | 34.9 ± 29.8<br>43.5 ± 23.8 | 3.38 to 30.4/13.6<br>7.56 to 36.6/20.7 | 13.9 ± 6.24<br>20.3 ± 9.73 | 5.02 to 134/33.9<br>19.3 to 80.1/38.3 | 39.4 ± 33.0<br>41.6 ± 21.7 | 14.1 to 170/42.4<br>38.9 to 103/54.2 | 55.1 ± 34.9<br>62.2 ± 23.2 | 9.15 to 67.5/23.6<br>29.2 to 78.5/48.6 | 28.4 ± 14.5<br>49.8 ± 17.8 |

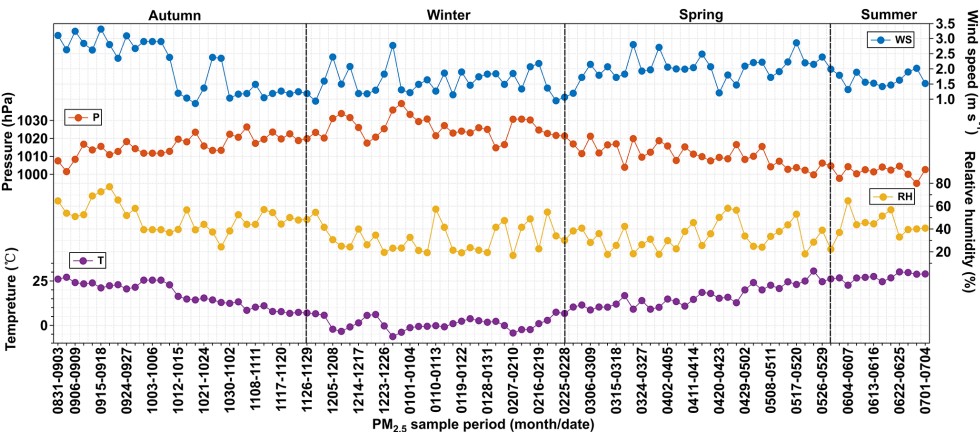

**Figure 2.** Temporal variations of ambient temperature ($T$), atmospheric pressure ($P$), wind speed (WS) and relative humidity (RH) in Tianjin from September 2018 to July 2019.

forcement of the control and prevention measures on pollutant emissions from various sources to improve the air quality further in this region.

## 3.3 Concentrations and seasonal variations of carbonaceous components

On average, the OC accounted for 17.3 % in $PM_{2.5}$ mass at ND and 13.3 % at HEP in Tianjin. The relative abundance of WSOC in OC was found to be $71.1 \pm 14.5$ % at ND and $65.6 \pm 16.8$ % at HEP. The average concentration of SOC was $3.11 \pm 3.42 \, \mu g \, m^{-3}$ at ND and $3.76 \pm 3.44 \, \mu g \, m^{-3}$ at HEP, accounting for 53.3 % and 57.5 % in OC, respectively. OC, WSOC and SOC showed clear seasonal changes during the campaign (Fig. 4). At ND, the average concentrations of OC and WSOC were higher in winter than in autumn, followed by spring and summer. On average, OC was 4 times higher in winter than that in summer at both ND and HEP. However, the average concentration of EC in winter was only about 1.7 times compared to that in summer at ND and 1.3 times at HEP. The higher loading of OC compared to that of EC in winter indicates that the OC emission from coal/biomass combustion should have been higher rather than EC in winter. In addition, the secondary formation of OC might be significant via adsorption and/or $NO_3$ radical-driven oxidation reactions of VOCs (G. Wang et al., 2016; Robinson et al., 2007). On the other hand, the temperate continental climate prevails over the Tianjin region, and the East Asian monsoon brings the humid oceanic air masses during summer that result in frequent precipitation events, which might cause the enhanced wet deposition of pollutants, including EC in summer (Y. Wang et al., 2016b; Luo et al., 2018; Tao et al., 2014). Interestingly, the average concentration of SOC in winter ($6.78 \, \mu g \, m^{-3}$, ND; $8.68 \, \mu g \, m^{-3}$, HEP) was found to be 6 times higher than that in summer ($1.08 \, \mu g \, m^{-3}$, ND; $1.18 \, \mu g \, m^{-3}$, HEP), which indicates that the formation of SOC was highly significant in the Tianjin

atmosphere during winter. The average WSOC was 0.69–16.0 at ND and 0.66–9.44 $\mu g \, m^{-3}$ at HEP. Such a higher level of WSOC at ND compared to that at HEP indicates its higher emission (potentially from biomass burning) and/or secondary formation under a high abundance of oxidants at ND than that at HEP.

## 3.4 Implications for $PM_{2.5}$ sources through relationships and mass ratios of carbonaceous components

Generally, EC does not react at ambient temperature and remains relatively stable in the atmosphere, and hence it is often used as a tracer for primary pollutants. Therefore, the scatter plots between EC and OC and their mass ratios can provide insights in tracing the origins of atmospheric aerosols and the extent of secondary formation of OC in the atmosphere. As shown in Fig. 5, OC showed a moderate correlation with EC in $PM_{2.5}$ at ND in spring ($R^2 = 0.45$, $p < 0.05$), summer ($R^2 = 0.50$, $p < 0.05$) and winter ($R^2 = 0.54$, $p < 0.05$) but a weak correlation ($R^2 = 0.05$, $p < 0.05$) in autumn. Such linear relations suggest that both OC and EC might have been derived from similar sources in spring, summer and winter at ND, whereas their sources might be different in autumn. The slope value was found to be higher in winter, which indicates that the contribution of OC from primary sources was higher in winter than in the other seasons. However, at HEP, the correlation between OC and EC in $PM_{2.5}$ in spring, summer, autumn as well as winter was very poor (Fig. 5), which implies that the sources of OC and EC were significantly different at HEP. Such differences between ND and HEP suggest that possible emissions of biogenic VOCs from rich vegetation including agricultural plants and/or biomass burning might be high at HEP and in the surrounding areas, and those VOCs must be subjected to in situ photochemical oxidation, resulting in high loading of OC compared to that of EC. As can be seen from Fig. 5d–e, $PM_{2.5}$ showed a high correlation with OC in autumn, winter

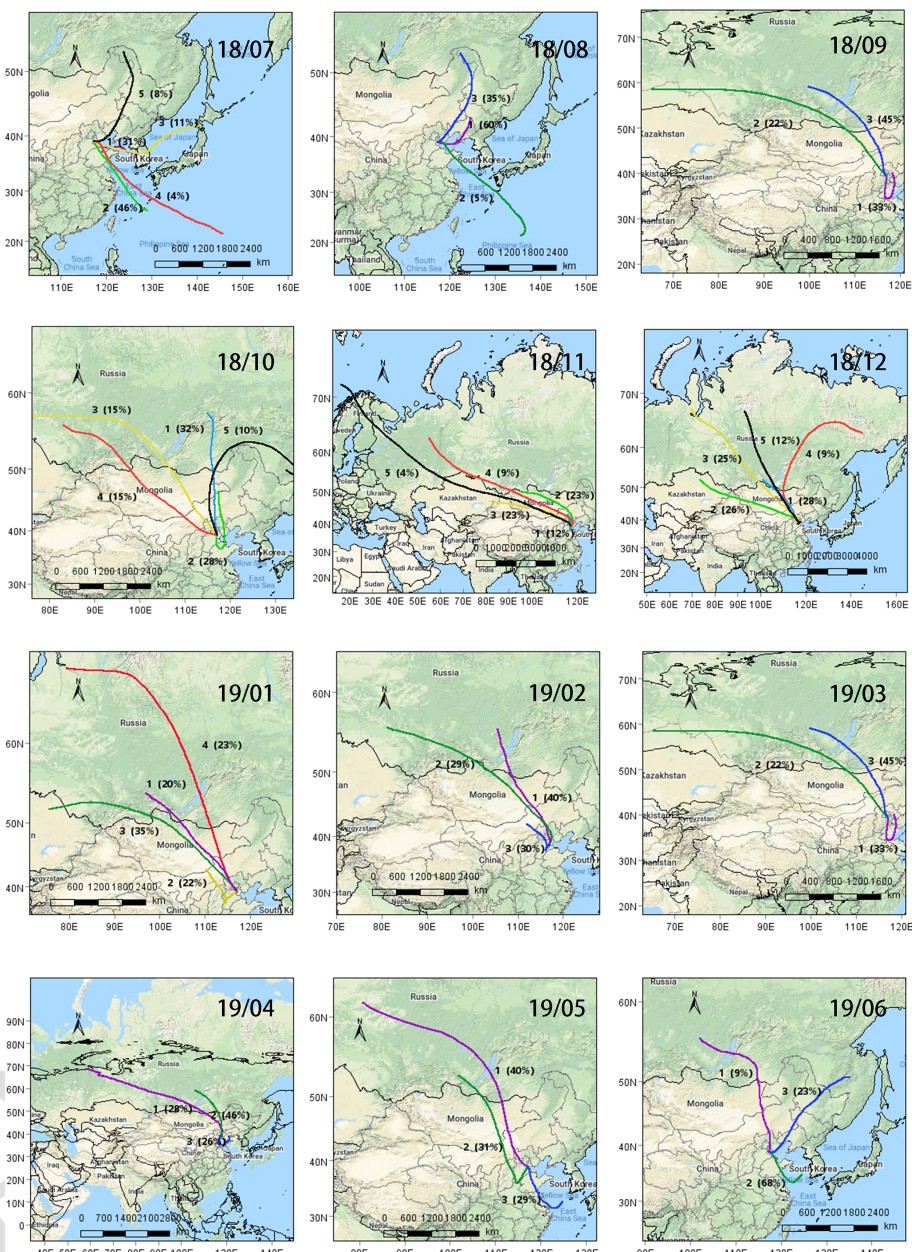

**Figure 3.** Monthly cluster analysis plots of 5 d backward air mass trajectories arriving over Tianjin at 500 m a.g.l. during the campaign period. (The maps were generated by the MeteoInfo software using © Google Maps.)

and spring at both the sampling points but a very poor correlation with EC, except in summer at HEP, which indicates that the loading of PM$_{2.5}$ was mainly driven by the OC loading. However, at HEP, the PM$_{2.5}$ showed a good correlation with EC in winter (Fig. 5e), indicating that the contribution from the primary sources was also important at HEP in winter.

Generally, the OC/EC ratio in the atmosphere is used to identify the emission and transformation characteristics of carbon particles. Chow et al. (2007) reported that when the OC/EC is higher than 2.0, it could be considered that the sec-

ondary formation of OC in the atmosphere is significant. On the other hand, the OC/EC varies significantly depending on their relative contributions from the emissions of coal combustion (range: 8.1–12.7), vehicle exhaust (0.7–2.4), biomass burning (4.1–14.5), wood combustion (16.8–40.0) and cooking (32.9–81.6) (Watson et al., 2001). The OC/EC ratios were 6.56–48.1 at ND and 4.01–63.0 at HEP, which are close to those reported for the particles emitted from biomass burning, including wood combustion and coal combustion, but not those from diesel and gasoline-driven vehicle exhaust.

**Table 2.** PM$_{2.5}$ mass concentrations in Tianjin and those reported at other different locales in China and around the world.

| City/nation | Sampling period | PM$_{2.5}$ (µg m$^{-3}$) | Reference |
| --- | --- | --- | --- |
| Tianjin, North China (urban site) | 2018–2019 | 34.9 ± 29.7 | This study |
| Tianjin, North China (suburban petrochemical industrial site) | 2018–2019 | 43.47 ± 23.5 | This study |
| Zibo, North China | 2006–2007 | 164.61 ± 79.14 | Luo et al. (2018) |
| Beijing, North China | 2009–2010 | 135.0 | Zhang et al. (2013) |
| Beijing, North China | 2013 | 84 | Xu and Zhang (2020) |
| Beijing, North China | 2018 | 50 | Xu and Zhang (2020) |
| Tianjin, North China | 2008 | 109.8 | Gu et al. (2010) |
| Harbin, Northeast China | 2017 | 59.39 ± 46.9 | Chen et al. (2019) |
| Chengdu, Southeast China | 2017 | 56.3 ± 28.1 | Huang et al. (2018) |
| Chengdu, Southeast China | 2014–2015 | 67.0 ± 43.4 | Wang et al. (2018) |
| Chengdu, Southeast China | 2011 | 119 ± 56 | Tao et al. (2014) |
| Ningbo, coastal Southeast China | 2012–2013 | 42.4 | M. Li et al. (2017) |
| Nanjing, coastal Southeast China | 2013–2014 | 129 | Li et al. (2016) |
| Guangzhou, South China | 2012-2013 | 44.2 | Lai et al. (2016) |
| Shanghai, coastal Southeast China | 2011–2012 | 68.4 | Zhao et al. (2015) |
| Los Angeles, USA | 2005–2006 | 19.88 | Hasheminassab et al. (2014) |
| Atlanta–Yorkville, USA | 2001–2005 | 14.3 | Chen et al. (2012) |

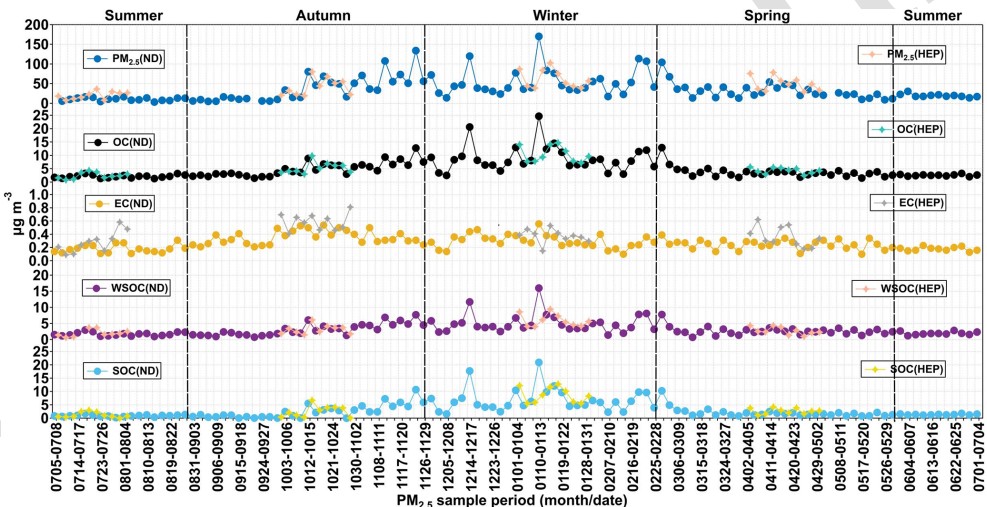

**Figure 4.** Temporal variations in concentrations (µg m$^{-3}$) of PM$_{2.5}$ and its chemical components OC, EC, WSOC and SOC at ND (solid dots) and HEP (solid stars) in Tianjin during 2018–2019. See the text for abbreviations.

The average OC/EC was 17.8 at ND and 15.7 at HEP, which are 8.7 and 7.8 times higher than 2.0, which indicates that the significant secondary formation of OA over the Tianjin region was significant. It has been reported that the ambient OC/EC in aerosols was gradually increasing over the period from 2000 to 2010 in China, confirming the increase in OC that should have been produced by enhanced oxidation in the atmosphere rather than from primary emissions (Cui et al., 2015). The high OC/EC ratios in the atmosphere of Tianjin once again demonstrated the enhanced emission and/or secondary formation of OC in China. On the other hand, the OC/EC ratio in winter was significantly higher than that in other seasons, especially at HEP, despite the fact that it is a suburban area (Fig. 6). Therefore, the increase in OC/EC in winter can be attributed to the increase in emission of VOCs from coal combustion due to its enhanced consumption for domestic heating and subsequent secondary formation of OC under stagnant weather conditions.

WSOC is mainly generated by oxidation reactions of VOCs in the atmosphere in addition to primary emissions such as biomass burning, and hence the mass fraction of WSOC in OC has been considered to be an indicator of aging of aerosols in the atmosphere, when the contribution of the WSOC is relatively low or insignificant from the biomass burning emission (Aggarwal and Kawamura, 2009). As shown in Fig. 5f, the correlation between PM$_{2.5}$ and

WSOC was much lower in summer than that in other seasons, which indicates that the secondary formation of the WSOC was more important than its primary emission, particularly in summer. Interestingly, WSOC/OC in Tianjin aerosols was found to be higher in spring and summer than in winter and autumn at both sites (Fig. 6). Such a high abundance of WSOC indicates the enhanced secondary formation of OC in spring and summer rather than in autumn and winter, because the biomass/biofuel combustion is significantly lower and because the VOC emissions from marine and terrestrial biota including croplands and subsequent photochemical processing are intensive in the gas and/or aqueous phases in spring/-summer (Padhy and Varshney, 2005; Pavuluri et al., 2013) compared to that in autumn/winter. In fact, being a coastal city, Tianjin receives marine air masses that are enriched with marine biological emissions due to the occurrence of sea breeze during daytime that are subjected to subsequent photochemical oxidation in the atmosphere. In addition, the air masses arriving in Tianjin during summer originated from the oceanic region (Fig. 3) and were also enriched with marine biological emissions and aged during the long-range atmospheric transport. On the other hand, the range and average WSOC/OC in Tianjin aerosols at ND and HEP (Table 1) are similar to those reported at urban locations, Nanjing, China (0.40–0.51) (Yang et al., 2005), and Chennai, India (0.23–0.61) (Pavuluri et al., 2011), and in largely rural areas of Hungary (range 0.38–0.72, average 0.66) (Kiss et al., 2002), where biomass burning was considered to be the major source of aerosols and aged during long-range atmospheric transport. Furthermore, WSOC and OC showed a very good linear relationship at both sites in all the seasons, which indicates that the contribution of OA from biomass burning emissions was also significant in addition to the secondary formation and/or transformations, particularly in autumn and winter, in the Tianjin region.

Interestingly, SOC/OC ratios were found to be higher in winter, followed by spring, summer and autumn (Table 1). The higher loading of SOC in winter might have occurred due to enhanced absorption/adsorption of VOCs to existing particles. In addition, despite lower temperatures prevailing over the Tianjin region, the secondary formation of OA might be intensive in winter by $NO_3$ radical reactions. It has been reported that the haze formation in China is mainly driven by the enhanced secondary formation of aerosols by $NO_3$ radical reactions (J. Wang et al., 2016). In fact, average concentrations of $NO_2$ that can be oxidized to $NO_3$ radicals by $O_3$ (Brown and Stutz, 2012) were significantly higher in winter (54.4 $\mu g\,m^{-3}$), followed by autumn (52.3 $\mu g\,m^{-3}$) and spring (39.1 $\mu g\,m^{-3}$), and lower in summer (30.3 $\mu g\,m^{-3}$) in the Tianjin atmosphere during the campaign (Li et al., 2021). Such higher levels of $NO_2$ might substantially be transformed to $NO_3$ radicals and accelerate the oxidation of VOCs of mostly biogenic origin, preferably through unimolecular reactions (Draper et al., 2019; Ng et al., 2017), and thus promote the formation of SOA, including organic

nitrates, which may not be fully water-soluble. In fact, the average concentration of WSOC was higher than that of SOC in spring, summer and autumn but opposite in winter (Table 1). Such differences indicate that the SOC produced in spring, summer and autumn might be mostly water-soluble, whereas in winter, part of the SOC is water-insoluble. In fact, WIOC accounts for 41.8 % of OC in winter at ND and 43.2 % at HEP, suggesting that part of the SOC (e.g., N-containing organics) might be water-insoluble. However, the temporal trends of WIOC, SOC and WSOC were similar, which implies that they should have originated from the same/similar sources and that their formation processes might also be similar in each season over the Tianjin region.

Furthermore, SOC showed a strong correlation with WIOC at both ND and HEP ($R^2 = 0.86$, $p < 0.05$ and 0.67, $p < 0.05$), and their slope values were significantly higher in winter but not in summer ($R^2 = 0.05$, $p < 0.05$ and 0.00, $p < 0.05$; Fig. 5). Such differences clearly imply that the secondary formation and/or transformation processes were quite different in winter from those in summer, and most of the SOC generated in winter was water-insoluble. Simulations, field observations, and laboratory studies have shown that the secondary formation of OA in the atmosphere over China is enhanced in winter, and only the aqueous-phase secondary formation has been considered the prominent pathway (Huang et al., 2014; J. Wang et al., 2016). Therefore, the enhanced formation of SOC in Tianjin aerosols, including WIOC, warrants further investigation of the possible formation processes of the WIOC, particularly in winter under the high abundance of $NO_3^-$, a subject of further research.

## 3.5   Implications for PM$_{2.5}$ sources through $\delta^{13}C_{TC}$

The box-and-whisker plots of seasons and annual $\delta^{13}C_{TC}$ and $\delta^{15}N_{TN}$ in Tianjin aerosols are depicted in Fig. 7. The $\delta^{13}C_{TC}$ was $-26.5\,‰$ to $-21.9\,‰$ with an average of $-25.0 \pm 0.7\,‰$ at ND (Table 1). They showed a temporal trend with a gradual enrichment of $^{13}$C in autumn and winter followed by a gradual depletion in the $^{13}$C to early summer and remained stable thereafter, except for a few cases at ND (Fig. 8), while at HEP, $\delta^{13}C_{TC}$ $-25.5\,‰$ to $-22.8\,‰$ (average $-24.5 \pm 0.55\,‰$) during the campaign period and their seasonal variations were similar to those found at ND, which indicates that the Tianjin aerosols should have been significantly derived from different sources in different seasons. The decreasing trend of $\delta^{13}C_{TC}$ from late winter to early summer through spring confirms the important role of biological emissions, because the VOCs and unsaturated fatty acids emitted from higher plants are depleted in $^{13}$C, as evidenced by the $\delta^{13}$C of fatty acids in unburned $C_3$ vegetation (range: $-38.5\,\%$ to $-32.4\,\%$) (Ballentine et al., 1998). In summer, the stability of $\delta^{13}C_{TC}$ might have been controlled by significant aging of OA under high solar radiation through enhanced photochemical reactions, which simultaneously led to the enrichment of $^{13}$C in reactants and

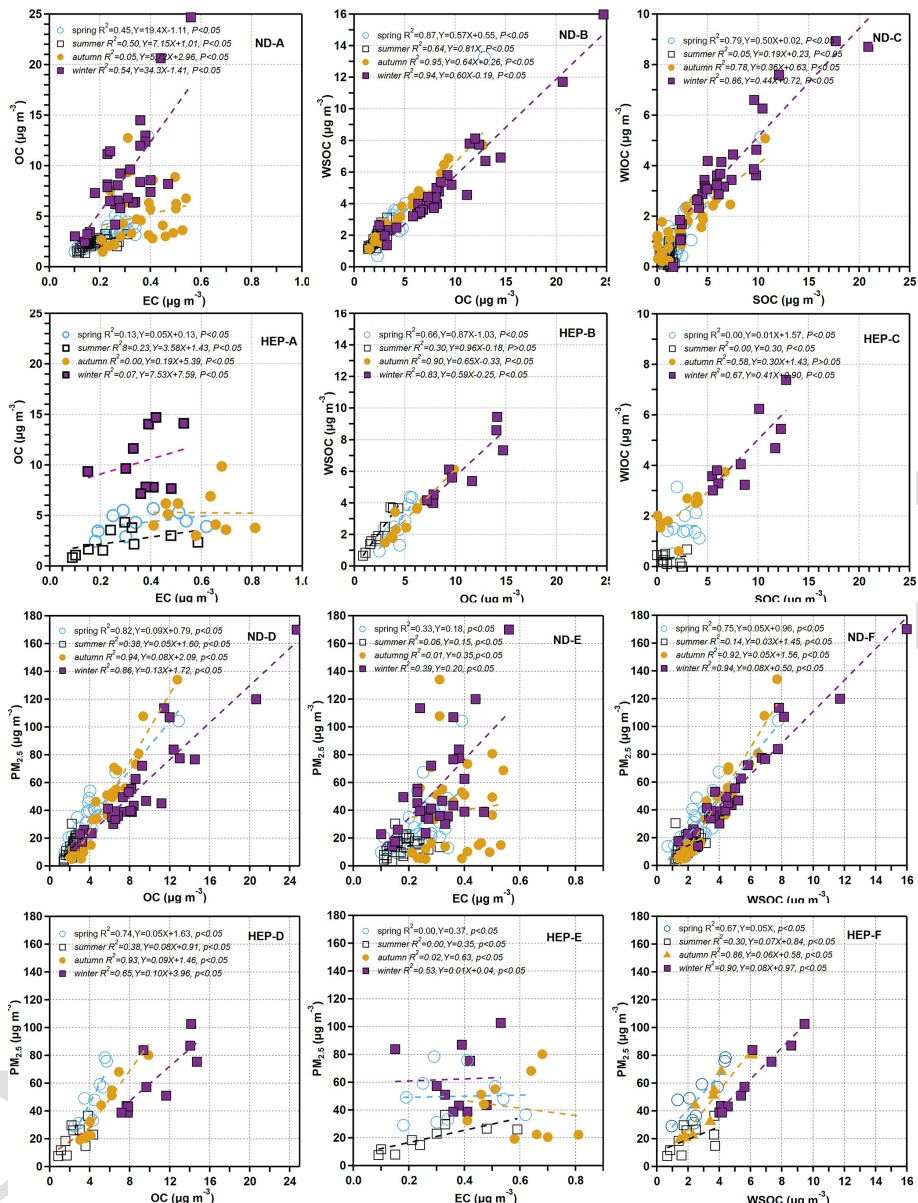

**Figure 5.** Scatter plots of selected carbonaceous components in PM$_{2.5}$ in Tianjin at ND and HEP.

its depletion in product compounds. The increasing trend of $\delta^{13}C_{TC}$ in autumn and winter indicates that the contribution of carbonaceous aerosols from biomass burning and fossil fuel combustion was large. The enrichment of $^{13}$C occurred in particles produced by biomass burning, while the $\delta^{13}$C of aerosol carbon produced by fossil fuel combustion was relatively higher than that of aerosol carbon produced by biological sources. In fact, the consumption of fossil fuels for heating in winter in Tianjin is much higher than in other seasons.

Figure 9 shows the $\delta^{13}C_{TC}$ of the particles emitted from point sources and/or source materials reported in the literature together with those found in Tianjin aerosols at both ND and HEP. The average $\delta^{13}C_{TC}$ at ND was comparable to that reported for total suspended particles (TSPs) over the western South China Sea (SCS), which were considered to be significantly influenced by biomass burning emissions, especially C$_3$ plants (Song et al., 2018). They were also comparable to those reported in aerosols (fine mode ($D_P < 2$ μm) and PM$_{10}$) in the Santarem region and in Mumbai, India, where biomass/biofuel burning emissions were expected to be the major sources of carbonaceous aerosols (Cloern et al., 2002; Pavuluri et al., 2015c). Furthermore, the average $\delta^{13}C_{TC}$ in HEP aerosols was similar to that reported in TSPs from Mount Tai in early June, which was considered to be highly influenced by burning activities of crop residues in

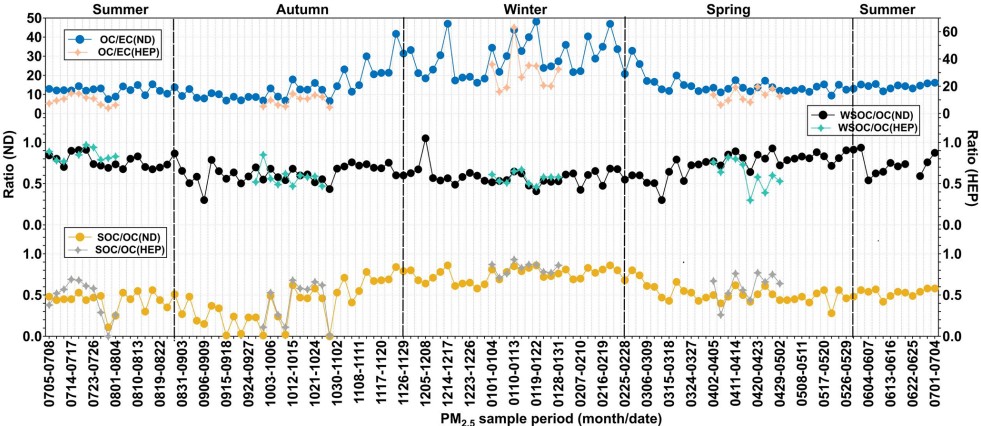

**Figure 6.** Temporal variations in the ratios of OC/EC, WSOC/OC and SOC/OC in PM$_{2.5}$ at ND and HEP in Tianjin during 2018–2019. See the text for abbreviations.

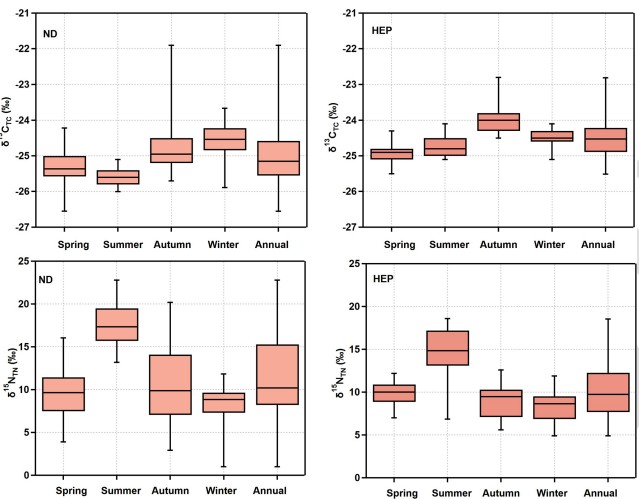

**Figure 7.** Box-and-whisker plot of seasonal variations in stable carbon and nitrogen isotope ratios of total carbon ($\delta^{13}$C$_{TC}$) and nitrogen ($\delta^{15}$N$_{TN}$) in PM$_{2.5}$ at ND and HEP in Tianjin during the campaign. The cross bar in the box shows the median, and the open circles show the outliers.

the North China Plain (Fu et al., 2012). Pavuluri and Kawamura (2017) reported a similar $\delta^{13}$C$_{TC}$ ($-24.8 \pm 0.68$‰) in TSPs in Sapporo, which were also strongly influenced by biomass burning and fossil fuel combustion emissions. Such comparisons clearly imply that the biomass burning emissions are the major sources of atmospheric aerosols in the Tianjin region, although we do not preclude the importance of other sources.

### 3.6    Concentrations and seasonal variations of nitrogenous components and other inorganic ions

Concentrations of the measured water-soluble inorganic ions showed the high abundance

of NO$_3^-$ at both the sites, followed by NH$_4^+$ > SO$_4^{2-}$ > Cl$^-$ > K$^+$ > Na$^+$ > Ca$^{2+}$ > Mg$^{2+}$ at ND and SO$_4^{2-}$ > NH$_4^+$ > Cl$^-$ > K$^+$ > Ca$^{2+}$ > Na$^+$ > Mg$^{2+}$ at HEP in Tianjin. Averages of the sums of ions were $18.7 \pm 16.9\,\mu\mathrm{g\,m^{-3}}$ and $22.7 \pm 13.1\,\mu\mathrm{g\,m^{-3}}$ (Table 1), accounting for 55 % of the PM$_{2.5}$ mass at ND and 56 % at HEP. SO$_4^{2-}$, NO$_3^-$ and NH$_4^+$ were found to be the major ions, and their total concentrations accounted for 89 % in the total concentration of the measured ions at ND and 87 % at HEP. Among them, SO$_4^{2-}$, NO$_3^-$ and NH$_4^+$ were 33 %, 31 % and 25 %, respectively, at ND and 29 %, 33 % and 24 %, respectively, at HEP. The concentration of NO$_3^-$ was the highest, accounting for 17 % of the PM$_{2.5}$ mass at ND and 18 % at HEP.

As can be seen from Fig. 10, the concentration of NO$_3^-$ peaked in winter and was lower in summer. In addition to primary emissions contributing a large amount of NO$_3^-$ in winter, it is likely because the low temperatures in winter promote the partitioning of NO$_3^-$ from the gas to particulate phases, whereas in summer, the higher temperatures enhance the transformation of NH$_4$NO$_3$ to HNO$_3$ (Utsunomiya and Wakamatsu, 1996), and the frequent precipitation might cause the wet deposition of the NO$_3^-$. The highest concentration of SO$_4^{2-}$ appeared in winter and the lowest in spring (Table 1). In winter, SO$_2$ emission is significantly increased due to the high consumption of fossil fuel, including coal combustion for domestic heating in the northern parts of China. The higher concentration of SO$_4^{2-}$ in summer than in spring might be due to higher temperature, relative humidity and abundant sunlight, which provide favorable conditions for the photochemical conversion of SO$_2$ to SO$_4^{2-}$ through gas- and aqueous-phase reactions. In addition, the air mass from the ocean in summer brings abundant SO$_4^{2-}$ from the perspective of emission sources. Interestingly, the seasonal variations of SO$_4^{2-}$ at HEP were quite different from those at ND, with the highest in summer and the lowest in autumn.

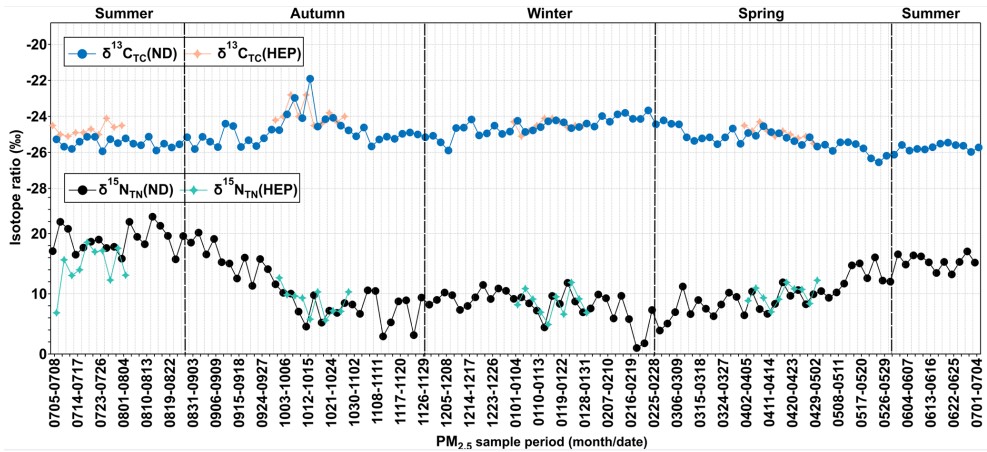

**Figure 8.** Temporal variations in $\delta^{13}C_{TC}$ and $\delta^{15}N_{TN}$ in PM$_{2.5}$ at ND (solid dots) and HEP (solid stars) in Tianjin during the campaign period (2018–2019).

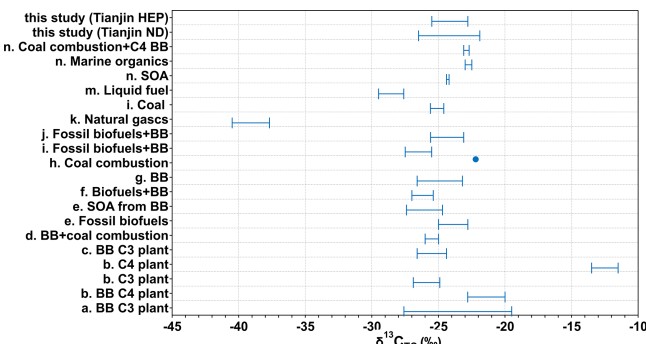

**Figure 9.** Range or mean $\delta^{13}C_{TC}$ in the particles emitted from point sources, source substances and atmospheric aerosols from different sites around the world. **(a)** Cao et al. (2016); **(b)** Martinelli et al. (2002); **(c)** Song et al. (2018); **(d)** Garbaras et al. (2015); **(e)** Bikkina et al. (2016); **(f)** Aggarwal et al. (2013); **(g)** Fu et al. (2012); **(h)** Kunwar et al. (2016); **(i)** Cachier et al. (1986); **(j)** Pavuluri and Kawamura (2016); **(k, l, m)** Widory (2007); **(n)** Kundu et al. (2014).

In addition, the loading of SO$_4^{2-}$ was always higher at HEP than at ND. In fact, as noted earlier, HEP was much closer to the seashore, and the aerosol composition must be more influenced by sea breeze during daytime throughout the year (Bei et al., 2018), while in summer, the air masses originated from the oceanic region that should have been enriched with marine biogenic emissions including dimethyl sulfide (DMS), which converts to SO$_2$ and then SO$_4^{2-}$ upon photochemical oxidation (Yan et al., 2020). On the other hand, the industries including petrochemical processing units are located near the seashore, and their emissions including SO$_2$ might have a significant impact on the aerosol composition at HEP, whereas at ND, local anthropogenic emissions, e.g., automobile exhausts, might have a greater influence on the composition of PM$_{2.5}$.

Since WSTN is mainly composed of IN ($\Sigma$NO$_3^-$–N + NH$_4^+$–N), the temporal trend of WSTN was found to be similar to that of IN (Fig. 11). Average concentrations of WSTN and IN were high at ND from mid-autumn to winter, and the IN peaked in mid-winter, whereas WSON peaked in late autumn. In addition, the average concentration of WSON was higher in autumn, followed by spring, winter and summer. On average, the mass fraction of WSON in WSTN was $6.74 \pm 7.81\%$ (range 0%–39.5%). At HEP, the average concentration of WSTN was $7.34 \pm 5.13\,\mu g\,m^{-3}$, and IN was $6.14 \pm 3.90\,\mu g\,m^{-3}$ (Table 1). Their average concentrations showed a seasonal pattern with higher levels in winter followed by spring and autumn, and the WSTN peaked in winter, whereas IN peaked in spring. In addition, the concentration of WSON was higher ($2.01 \pm 1.80\,\mu g\,m^{-3}$) in the growing season than that in winter and autumn.

## 3.7 Implications for PM$_{2.5}$ sources through mass ratios and relationships of nitrogenous components and other inorganic ions

The mass ratio of NO$_3^-$ to SO$_4^{2-}$ reflects the relative contribution from local moving sources (motor vehicles) and fixed sources (including coal combustion) to atmospheric aerosols. Generally, if the ratio is $\geq 1$, automobile exhaust is considered to be an important source of the particles in the given environment (Ming et al., 2017). The NO$_3^-$/SO$_4^{2-}$ ratio was found to be higher than 1 in all the seasons, except for summer (0.21), and the annual average was 1.63 at ND, which indicates that the automobile exhaust was also an important source of the PM$_{2.5}$ in the urban area of Tianjin. In summer, the air masses originating from the oceanic region should have been enriched with the marine biogenic emissions including DMS, and thus the contribution of biogenic SO$_4^{2-}$ might be significant in Tianjin aerosol. The NO$_3^-$/SO$_4^{2-}$ in Tianjin (ND: 1.63, HEP: 1.35) is similar to that reported in

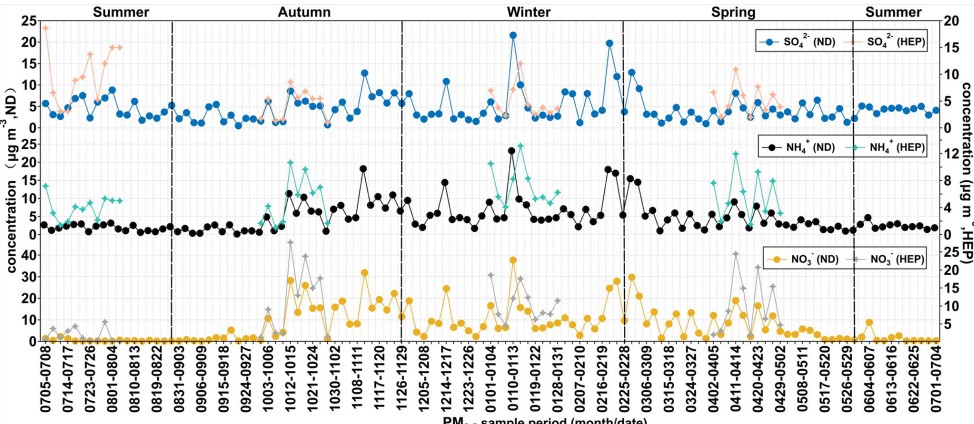

**Figure 10.** Temporal variations in concentrations ($\mu g\,m^{-3}$) of secondary ionic species in PM$_{2.5}$ at ND (solid dots) and HEP (solid stars) in Tianjin during the campaign period (2018–2019).

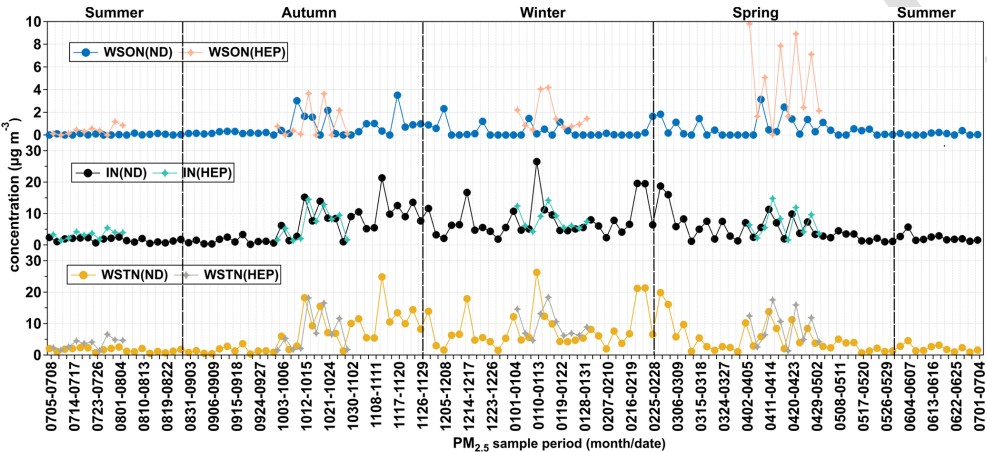

**Figure 11.** Temporal variations in WSTN, IN and WSON ($\mu g\,m^{-3}$) in PM$_{2.5}$ at ND (solid dots) and HEP (solid stars) in Tianjin during the campaign period (2018–2019).

Beijing (1.37) (Xu et al., 2017) and Shanghai (1.05), where automobile exhausts have been considered one of the major sources. Such comparability again supports our finding that automobile exhausts are an important source of aerosols in Tianjin.

The correlation between $SO_4^{2-}$ and $NO_3^-$ was good in spring ($R^2 \geq 0.55$, $p = 0.08$), autumn and winter ($R^2 \geq 0.55$, $p < 0.05$) but not in summer ($R^2 = 0.00$, $p < 0.05$ at ND and 0.06, $p < 0.05$ at HEP). Such comparability might appear to be due to high emissions of $NO_x$ and $SO_2$ from fossil fuel, including coal combustion during the cold period (late autumn to the following early spring) and subsequent secondary formation, whereas in summer, the emission of $SO_2$ from coal combustion in the industrial sector and marine biogenic emission of DMS might be larger than in other seasons. In addition, the $NH_4NO_3$ is more susceptible to the decomposition into gaseous $HNO_3$ and $NH_3$ at higher temperatures (Russell et al., 1983) that prevailed in summer. The an-

nual average concentration of $Cl^-$ was $1.45 \pm 1.79\,\mu g\,m^{-3}$, accounting for 4.15 % of the PM$_{2.5}$ mass at ND with the higher loading in winter than in the other seasons. Such a high loading again confirms the enhanced consumption of coal in winter for domestic heating, because the emission of $Cl^-$ is abundant from coal combustion (Zhang et al., 2017; He et al., 2001), while $K^+$ was also found to be higher in winter, followed by autumn, spring and summer (Table 1). The high loading of $K^+$ in winter might be due to enhanced biomass burning for domestic heating.

$SO_4^{2-}$ and $NO_3^-$ showed a good correlation with $NH_4^+$ at ND and moderate and good correlations at HEP but weak or no correlation with alkali ($Na^+$, $Ca^{2+}$ and $Mg^{2+}$) ions at both sites (Table 3), suggesting that they were mainly associated with $NH_4^+$ in the form of $(NH_4)_2SO_4/NH_4HSO_4$ and $NH_4NO_3$ rather than with alkali metals. Interestingly, the $SO_4^{2-}$, $NO_3^-$ and $NH_4^+$ showed a medium correlation with $K^+$, except for $SO_4^{2-}$ at HEP, which suggests that $SO_4^{2-}$,

**Table 3.** Correlation coefficients ($R^2$) of inorganic ions in PM$_{2.5}$ at ND (right) and HEP (left) in Tianjin, North China.

| | PM$_{2.5}$ | Cl$^-$ | SO$_4^{2-}$ | NO$_3^-$ | Na$^+$ | NH$_4^+$ | K$^+$ | Mg$^{2+}$ | Ca$^{2+}$ |
|---|---|---|---|---|---|---|---|---|---|
| PM$_{2.5}$ | | 0.71 | 0.63 | 0.86 | 0.29 | 0.90 | 0.64 | 0.22 | 0.26 |
| Cl$^-$ | 0.43 | | 0.31 | 0.56 | 0.40 | 0.62 | 0.67 | 0.27 | 0.24 |
| SO$_4^{2-}$ | 0.05 | 0.03 | | 0.54 | 0.10 | 0.74 | 0.46 | 0.16 | 0.09 |
| NO$_3^-$ | 0.57 | 0.21 | 0.03 | | 0.21 | 0.92 | 0.49 | 0.11 | 0.15 |
| Na$^+$ | 0.17 | 0.07 | 0.09 | 0.13 | | 0.21 | 0.20 | 0.05 | 0.19 |
| NH$_4^+$ | 0.80 | 0.30 | 0.24 | 0.71 | 0.18 | | 0.56 | 0.15 | 0.16 |
| K$^+$ | 0.64 | 0.71 | 0.00 | 0.47 | 0.19 | 0.55 | | 0.66 | 0.24 |
| Mg$^{2+}$ | 0.36 | 0.30 | 0.00 | 0.04 | 0.02 | 0.21 | 0.27 | | 0.45 |
| Ca$^{2+}$ | 0.38 | 0.13 | 0.01 | 0.07 | 0.06 | 0.25 | 0.18 | 0.81 | |

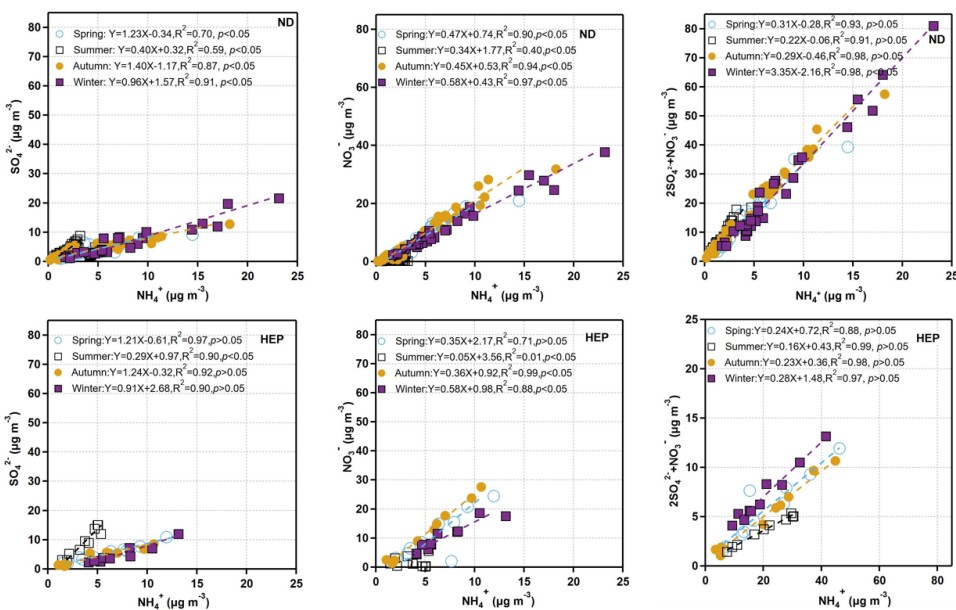

**Figure 12.** Linear relations between secondary ions in PM$_{2.5}$ at ND and HEP in Tianjin during the campaign period.

NO$_3^-$ and NH$_4^+$ might have been significantly derived from biomass burning emissions. The nonexistent correlation between K$^+$ and SO$_4^{2-}$ indicates that the sources of SO$_4^{2-}$ were significantly different from those of NH$_4^+$, NO$_3^-$ and K$^+$ at HEP. As discussed in the previous section, the SO$_4^{2-}$ might have been significantly derived from industrial emissions, particularly from petrochemical plants that existed near HEP, and/or a larger contribution of SO$_4^{2-}$ derived from marine biogenic emissions due to sea breeze. The correlation coefficient between Ca$^{2+}$ and Mg$^{2+}$ was relatively high (Table 3), which indicates that they might have been emitted from the same source such as soil dust. However, the mass ratio of Mg$^{2+}$ to Ca$^{2+}$ was 0.27 at ND and 0.14 at HEP, which is comparable to those reported at the point source of coal combustion (Wang et al., 2005), implying that the Ca$^{2+}$ and Mg$^{2+}$ in Tianjin aerosols are derived not only from soil dust, but also from coal combustion emissions.

The molar ratios of NH$_4^+$/SO$_4^{2-}$, NH$_4^+$/NO$_3^-$ and NH$_4^+$/(2SO$_4^{2-}$ + NO$_3^-$) can indicate their coexistence forms (Lyu et al., 2015; Behera et al., 2013). Figure 12 shows the linear relations between NH$_4^+$ and SO$_4^{2-}$, NO$_3^-$ and (2SO$_4^{2-}$ + NO$_3^-$). NH$_4^+$ showed significant correlations with SO$_4^{2-}$ and NO$_3^-$ except for summer, confirming that sufficient NH$_3$ was present to neutralize H$_2$SO$_4$ and HNO$_3$ during the campaign period. The relatively higher correlation of NH$_4^+$ with NO$_3^-$ than that with SO$_4^{2-}$ suggests that NH$_4$NO$_3$ might be more likely formed than (NH$_4$)$_2$SO$_4$ because of a better affinity between the two ions (Blanchard and Hidy, 2003) at both sites (Table 3). Furthermore, the slopes and coefficients between the selected ions (Fig. 12) indicated that NH$_4$NO$_3$, (NH$_4$)$_2$SO$_4$, NH$_4$HSO$_4$ and NH$_4$NO$_3$ were the more likely existing forms of secondary inorganic ions in Tianjin in all seasons, except for summer, during which the (NH$_4$)$_2$SO$_4$

might have existed due to the loss of $HNO_3$ and enhancement of $NH_3$ emissions at high temperatures.

## 3.8 Implications for PM$_{2.5}$ sources through $\delta^{15}N_{TN}$

$\delta^{15}N_{TN}$ was 1.10‰–22.8‰ (11.4 ± 4.8‰) at ND and
4.91‰–18.6‰ (10.4 ± 3.4‰) at HEP during the campaign.
The temporal trends at ND and HEP were highly comparable with each other (Fig. 7). The averages of $\delta^{15}N$ varied significantly from season to season, with higher values in summer (17.7 ± 2.51‰ at ND and 14.5 ± 3.3‰ at HEP)
and lower values (8.07 ± 2.5‰ at ND and 8.41 ± 2.0‰ at HEP) in winter. Such seasonal changes in $\delta^{15}N_{TN}$ suggest that the aerosol N was significantly influenced by season-specific source(s) and/or the chemical aging of N species.

The range (or average) of $\delta^{15}N$ reported for the particles
emitted from point sources as well as those reported in atmospheric aerosols from different locations over the world together with those obtained in Tianjin PM$_{2.5}$ are depicted in Fig. 13. $\delta^{15}N_{TN}$ in Tianjin PM$_{2.5}$ is slightly higher than that (−19.4‰ to 15.4‰) reported for the particles emitted
from point sources of fossil fuel combustion and waste incineration burning (Fig. 13). They are also higher than those reported in the marine aerosols over the western North Pacific (4.9 ± 2.8‰), which were considered to be mainly derived from sea-to-air emissions (Miyazaki et al., 2011). How-
ever, $\delta^{15}N_{TN}$ in Tianjin PM$_{2.5}$ are comparable to the higher ends of the $\delta^{15}N_{TN}$ reported in atmospheric aerosols from Jeju Island, Korea (Fig. 13), which were attributed to vehicle emissions, coal burning and straw burning (Kundu et al., 2010), and to those reported in urban aerosols from Paris,
France, where fossil fuel combustion was expected to be a major source (Widory, 2007). Furthermore, the lower ends of $\delta^{15}N_{TN}$ in Tianjin PM$_{2.5}$ are comparable to the lower ends of $\delta^{15}N_{TN}$ reported for the particles emitted from the controlled burning of C$_3$ plant debris (range: +2.0‰ to +19.5‰)
(Fig. 13). The higher ends of $\delta^{15}N_{TN}$ in Tianjin PM$_{2.5}$ are comparable to the higher ends of $\delta^{15}N_{TN}$ from C$_4$ plant debris (+9.8‰ to +22.7‰) in a laboratory study and to those of atmospheric aerosols from Piracicaba and the Amazon basin, Brazil, where biomass burning is a dominant source
(Cloern et al., 2002) (Fig. 13). This is consistent with the fact that wheat and corn are the main crops in Tianjin. Such comparisons again confirm that biomass burning is a major source of atmospheric aerosols, followed by fossil fuel combustion in the Tianjin region.

## 4 Summary and conclusions

Fine aerosol (PM$_{2.5}$) samples were collected with a frequency of 3 consecutive days for each sample over a 1-year period from July 2018 to July 2019 at urban (ND) and suburban (HEP) sites in the Tianjin atmosphere, North China.
The PM$_{2.5}$ was studied for carbonaceous (OC, EC, WSOC, WIOC, SOC and TC), nitrogenous (WSTN, IN and WSON),

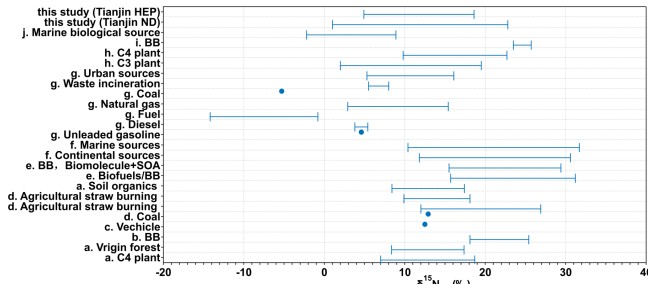

**Figure 13.** Range or mean $\delta^{15}N_{TN}$ in the particles emitted from point sources, source substances, and atmospheric aerosols from different sites around the world. **(a)** Martinelli et al. (2002); **(b)** Aggarwal et al. (2013); **(c)** Kunwar et al. (2016); **(d)** Kundu et al. (2010); **(e)** Pavuluri et al. (2010); **(f)** Bikkina et al. (2016); **(g)** Widory (2007); **(h)** Turekian et al. (1998); **(i)** Kundu et al. (2010); **(j)** Miyazaki et al. (2011).

and inorganic ionic ($Cl^-$, $NO_3^-$, $SO_4^{2-}$, $Na^+$, $K^+$, $NH_4^+$, $Ca^{2+}$ and $Mg^{2+}$) components as well as stable carbon and nitrogen isotope ratios of total carbon ($\delta^{13}C_{TC}$) and nitrogen ($\delta^{15}N_{TN}$). The characteristics of PM$_{2.5}$ and its components showed a clear seasonal pattern, with higher concentrations in winter and lower concentrations mostly in summer. The mass ratios of OC/EC, WSOC/OC and SOC/OC suggested that Tianjin aerosols were derived from coal combustion, biomass burning and photochemical reactions of VOCs and also implied that the Tianjin aerosols were more aged during long-range atmospheric transport in summer. The seasonal variation in ion concentrations highlighted that coal combustion was the main source of aerosol and that automobile exhaust also played an important role in controlling the Tianjin aerosol loading. In addition, the concentration of $SO_4^{2-}$ at HEP peaked in summer and was at its minimum in autumn, and the overall levels were higher at HEP than those at ND Tianjin, which suggested that the contribution of the marine air masses originated from the oceanic region in summer and sea breeze throughout the year, and industrial emissions, particularly from the petrochemical industry located at the seashore, were larger at HEP than at ND. The values of $\delta^{13}C_{TC}$ and $\delta^{15}N_{TN}$ also showed that biomass and coal combustion were the major sources of aerosols in autumn and winter, and dust, biological emissions and the oceanic emissions were major in spring and summer in Tianjin. Moreover, this study has also provided comprehensive baseline data on carbonaceous and inorganic aerosols as well as their isotope ratios over a 1-year period in the Tianjin region, North China.

**Data availability.** The data used in this study can be found online at https://doi.org/10.5281/zenodo.5140861 (Dong et al., 2021).

**Author contributions.** ZD and CMP conceptualized this study. ZD, YW and PL conducted the sampling. ZD conducted the chem-

ical analyses, interpreted the data and wrote the manuscript. CMP supervised the research and acquired the funding for this study. ZX administrated the project. CMP, ZX, PF and CQL contributed in discussing the results and review and editing the manuscript.

**Competing interests.** The contact author has declared that none of the authors has any competing interests.

**Disclaimer.** Publisher's note: Copernicus Publications remains neutral with regard to jurisdictional claims in published maps and institutional affiliations.

**Financial support.** This research has been supported by the National Key Research and Development Program of China (grant no. 2017YFC0212700) and the National Natural Science Foundation of China (grant no. 41775120).

**Review statement.** This paper was edited by Zhibin Wang and reviewed by two anonymous referees.

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

## Remarks from the typesetter

TS1    Please confirm.
TS2    The new URL has been inserted. It is still not resolving. Please check.
TS3    **Editor approval required**
TS4    **Editor approval required**
TS5    **Editor approval required**