# Peer review of "Measurement Report: Chemical components and 13C and 15N isotope ratios of fine aerosols over Tianjin, North China: Year-round observations"

_Atmospheric Chemistry and Physics, 2022_

## Author Comment (AC1)

**Authors' Response to Referee #1**

This manuscript entitled "Measurement report: ---- observations" by Dong et al. presents the comprehensive characterization and seasonality of carbonaceous, and nitrogenous components and inorganic ions as well as the stable carbon and nitrogen isotope ratios of total carbon and nitrogen in $PM_{2.5}$ collected continuously at an urban location and for one month in each season at a sub-urban location in Tianjin, north China over a one year period during 2018-2019. Overall, the data interpretation is logical and the paper is well written. Therefore, I recommend that this manuscript can be published after addressing the following minor remarks.

*Response:* We thank the referee for his/her critical reading of the manuscript, appreciation of our work and constructive comments and suggestions, which helped to improve the quality of the MS. The MS is revised according to all the comments from the referee. Our point-by-point responses to all the comments are provided below. Please see the revised MS for details of the revisions.

1. Typos and language errors need to be corrected throughout the text. For example: Abstract, L15: '--- water-soluble OC (WSOC ---- '. The bracket should be closed. L20: '---- winter, while biological and/or marine emissions ---' should be changed to "---winter, while terrestrial and/or marine biological emissions ---".

*Response:* We regret for the typos in the MS text. We took care to rectify all the language errors and typos throughout the text. Please check Lines 15 & 20 and other in the revised MS.

2. Introduction, L36~: I suggest the authors to introduce importance/impacts of specific (bulk) components, after the general introduction of aerosols.

*Response:* Following the reviewer's suggestion, we substantially improved the introduction on the importance of carbonaceous and reactive nitrogen components in detail in the revised MS (see Lines 40-48 and 75-81).

3. Aerosol sampling: Since each $PM_{2.5}$ sample was collected for relatively longer time (72 hrs), it is important to describe the potential sampling artefacts as well.

*Response:* Yes, we agree with the referee that there is a possibility of having sampling artefacts due the lengthy sampling period. We describe such possibility in detail in the revised MS (see Lines 140-148).

4. Figures 1 & other: I suggest the authors to depict the seasonal separation as well in the temporal variations, in order to provide the clear visibility to the reader, like in Figure 4.

*Response:* Following the referee's suggestion, we depicted the seasonal separation in all figures, which showing the temporal variations, in the revised MS.

5. L217-219 & 249-252 ..: Since the annual and seasonal data have been presented in Table 1, it is better to avoid noting the same repeatedly in text, rather referring the Table 1.

*Response:* Following the referee's suggestion, we removed most of the data presentation in the text in order to avoid such repetition in the revised MS.

---

## Author Comment (AC2)

This manuscript shows a detailed study on $PM_{2.5}$ in urban and suburban site of North China city (Tianjin). The study focused on the concentrations of different chemical components including carbonaceous (EC, OC, SOC, WSOC, WIOC, TC), nitrogenous (WSTN, IN, WSON) and other inorganic ions. Additionally, stable isotopes of total carbon and nitrogen in $PM_{2.5}$ were also shown. This sufficient and comprehensive study can help us further understand the source and atmospheric processes of fine aerosols in regional scale, and the data could help to promote scientific progress within the scope of Atmospheric Chemistry and Physics. However, quite a lot of necessary information that needed to help understanding the whole manuscript is lacking, and the paper is poorly written, the language and expressions need to be further improved. Detailed comments could be found as follows:

*Response:* We thank the referee for his/her critical reading of the manuscript, appreciation of our work and constructive comments and suggestions, which helped to improve the quality of the MS. The MS is revised according to all the comments from the referee. Our point-by-point responses to all the comments are provided below. Please see the revised MS for details of the revisions.

**Specific Comments:**

1. Major comments on introduction. The study aimed to explore the origins and atmospheric processes of fine particles through seasonal variations of carbonaceous (EC, OC, SOC, WSOC, WIOC, TC), nitrogenous (WSTN, IN, WSON), other inorganic ions and stable isotopes of TC and TN in urban and suburban site of Tianjin. Therefore, the background in introduction should include: why choose to study $PM_{2.5}$? why EC, OC, SOC, WSOC, WIOC, TC, WSTN, IN, WSON and stable isotopes are important in understanding the source and atmospheric process of aerosols? Why choose to study urban and suburban aerosols in Tianjin? Some of the information is presented in current version of the manuscript, however, more information needs to be added in introduction section. For example, the authors studied EC, OC, SOC, WSOC, WIOC, TC in the $PM_{2.5}$, however, there is only a simple introduction of EC and OC in the second paragraph, then why the authors also explored the seasonal variation of SOC, WSOC, WIOC? Are they important in understanding the source and atmospheric process of fine aerosols? Why? Similar problem also happens in nitrogenous components and other inorganic ions in introduction section. In addition, $\delta^{13}C_{TC}$ and $\delta^{15}N_{TN}$ of aerosols can be used to trace the emission source of aerosols, however, fractionation effects during the formation and transportation might modify the initial value of $\delta^{13}C$ and $\delta^{15}N$ from sources, which might lead to the uncertainties of directly using $\delta^{13}C$ and $\delta^{15}N$ in aerosols to trace source contributions. Therefore, the background about the role of fractionation effects in affecting the $\delta^{13}C$ and $\delta^{15}N$ in aerosols is important to understand the related result and its implications. However, no such information was found in current introduction section.

*Response:* Following the reviewer's comments/suggestions, we substantially improved the introduction section by adding the required contents on the importance of carbonaceous (EC, OC, WSOC, WIOC and SOC), nitrogenous (IN, ON and WSON) and inorganic ionic components in the revised MS (please see Lines 40-48 and 75-81).

Yes, we agree with the reviewer that the isotope fractionation during the secondary formation and transformation processes of aerosols modify the initial value of $\delta^{13}C$ and $\delta^{15}N$ of the aerosols. However, it would be significant in the case of $\delta^{13}C$ and $\delta^{15}N$ of molecular species, but relatively

lower in the case of the $\delta^{13}C$ and $\delta^{15}N$ of the total carbon (TC) and nitrogen (TN) unless gas-to-particle and/or particle-to-gas transitions are significant, because the TC and TN consist of both the reactants and products. Therefore, it is not possible to assess the influence of the isotope fractionation on the observed $\delta^{13}C$ and $\delta^{15}N$ of TC and TN, respectively, in $PM_{2.5}$. We noted this point in the revised MS (see Lines 88-95).

2. Major comments on Materials and methods. (1) Locations of the urban and suburban site needs to be indicated in a map to help better understanding of the results; (2) There are results of meteorology and backward air mass trajectories, however, no related information was found in Materials and methods section; (3) Necessary information is lacking. For example, what's the flow rate of the air sampler during sampling period? This is important, cause the authors continuously sampling for 72-h each time, if the flow rate is high, then I'm wondering whether the filter will be saturated or not, especially in winter when $PM_{2.5}$ is high; (3) Further explanation needs to be added to support the feasibility of the method. For example, the authors described "OC and EC were measured using OC/EC analyzer……, based on thermal light transmission ……and assuming the carbonate carbon was negligible." Why the carbonate carbon is negligible, is it really negligible in aerosols of Tianjin? In addition, the authors described that "The N contents of $NO_2^-$, $NO_3^-$ and $NH_4^+$ were calculated from their concentrations." but how? the authors need to explain more. Lastly, there are quite large uncertainties in WSTN, WSON etc., however, the authors consider "……such errors do not influence the conclusions drawn from this study.", why? explain more.

*Response:* Following the reviewer's suggestion, (1) We have added the map of China with the sampling points: urban and suburban areas, (2) noted the data source information of meteorological parameters and backward air mass trajectory in sub-section 2.3, and (3) noted the flow rate of the air sampler in Tianjin in the revised MS.

It has been reported that the C removed by HCl treatment accounted for only 6.3% in total carbon (TC) at Gosan Island, South Korea, where the long-range transported airmasses enriched with dust are the major sources, rather than anthropogenic sources (Kawamura et al., 2004). Whereas in Tianjin, the anthropogenic emissions and subsequent secondary processes are considered as the major sources, and the contribution of soil dust is relatively much lower. That is why, we assumed the carbonate carbon as negligible in this study. We noted this point in the revised MS (see Lines 156-159)

We added the computing method of the N contents of $NO_2^-$, $NO_3^-$ and $NH_4^+$ in the revised MS (see Line 201-202). As for the uncertainty of WSTN and WSON, we believe that this study explores the seasonal characteristics and possible sources of $PM_{2.5}$ in Tianjin, rather than the their atmospheric loadings. Since the uncertainty in the measurement of WSTN and WSON is common for samples, we believe that it may not seriously influence the overall conclusions.

3. Major comments on Results and discussion. The prominent problems in results and discussion are that (1) no statistic analysis of the results; (2) no literatures or data are provided to support the some of the explanations of the results. For example, in section 3.2, the authors expressed that "Furthermore, the average concentration of $PM_{2.5}$ found to be higher in spring than in autumn (Table 1), probably due to enhanced eruption of dust from open lands, due to gradual increase in wind speed in spring (Fig. 1).". First of all, the concentration of $PM_{2.5}$ is higher in spring than in autumn, is there any significant difference? Secondly, the authors owe this to "enhanced

eruption of dust from open lands", is there any reference to support this idea? For the other example, from lines 260-265, the authors said "…… the secondary formation of OC might be significant via adsorption and/or $NO_3$ radical driven oxidation reactions of VOCs." Are there any citations?? "…… the frequent precipitation events might result the enhanced wet deposition of……" Do you have any data about seasonal precipitation amount or reference to support this? These are only some examples chosen from the results and discussion section, in fact, there are quite a lot of sentences that need to be supported by reference. The authors need to carefully double check each sentence and complete with appropriate reference to confirm your conclusions.

*Response:* Following the reviewer's comments and suggestion, we thoroughly checked our interpretations and provided appropriate citations throughout the Discussion section in the revised MS. In fact, (1) we assessed the possible source of $PM_{2.5}$ based on the correlation between selected carbonaceous and ionic (marker) components and their statistical significance (p value). The p values are noted in the revised MS (see Section 3.3.1).

(2) Yes, the average concentration of $PM_{2.5}$ in spring is significantly higher than that in autumn at both sampling sites, which is twice higher than that in autumn. In fact, the dust storms over Mongolia and China are common in spring that enhance the loading of $PM_{2.5}$ in the East Asian atmosphere (Liu et al., 2011). We noted this point in the revised MS (see Lines 256-259).

(3) Secondary organic carbon (SOC) is generated from volatile organic compounds through physicochemical adsorption and photochemical reactions including multiphase reactions (Robinson et al., 2007; Wang et al., 2016). We cited these references in the revised MS (see Lines 293-295).

(4) Unfortunately, we do not have the rain data during the campaign. However, the temperate continental climate with high temperature prevails over the Tianjin region and the East Asian monsoon brings the humid oceanic air masses during summer that result in frequent precipitation events. Previous studies pointed out that more precipitation and stronger atmospheric vertical mixing in the summer was one of the reasons for the decrease of $PM_{2.5}$ concentration in summer (Wang et al., 2016;Luo et al., 2018;Tao et al., 2014). These points and citations are included in the revised MS (see Lines 295-297).

**Technical corrections:**

1. Line 42: Move "(2127 and 1356 Gg, respectively)" after "2000"; I addition, there are so many "respectively" through the whole manuscript, quite annoying and makes the sentences hard to understand. Generally, "respectively" is always used when to distinguish three or more different items, please double check and change the expressions through the manuscript.

*Response:* Following the reviewer's suggestion, we took care to avoid using the word "respectively" throughout the text in the revised MS.

2. Line 59, Please delete the "," after "thus".

*Response:* We deleted the "," mark in the revised MS (see Line 66).

3. Line 73, Change "theier" to "their".

*Response:* We corrected this typo in the revised MS (see Line 84).

4. Lines 90-91, Better give the area percentage of "agricultural fields and forests" around Tianjin.

*Response:* Following the reviewer's suggestion, we added the area information of agricultural fields and forests in order to express the effect of biogenic emissions on aerosols in Tianjin. (see Lines 105-109).

5. Lines 93-94, Still have no idea why Tianjin is the "ideal location".

*Response:* As detailed in Lines 103-112 in the revised MS, Tianjin receives the long-range transported air mass from different source regions (land and ocean), depending on season, in addition to the local anthropogenic emissions. That is why, Tianjin is considered as an ideal location for studying the aerosol characteristics of different origins and atmospheric processing (aging) in northern China.

6. Line 104, Change "measurement of its mass" to "mass measurement".

*Response:* Following the reviewer's suggestion, we modified it in the revised MS.

7. Line 131, Please explain "TIC, acidizing" and "wet oxidation".

*Response:* Following the reviewer's suggestion, we briefly described the methods of acidizing and wet oxidation of the sample in the revised MS (see Lines 165-167).

8. Line 173, There is a ", was 0.83"? What's that mean?

*Response:* The 0.83 is the propagation error of WSON with the repetitive errors of $NO_3^-$, $NH_4^+$ and WSTN, and doesn't possess any unit.

9. Lines 177-179, Such a long sentence, better break it into two or three sentences.

*Response:* Following the reviewer's suggestion, we divided this sentence into two in the revised MS (see Lines 212-214).

10. Lines 180-185, The final $\delta^{13}C$ and $\delta^{15}N$ is relative to VPDB and atmospheric $N_2$? Better make it clear.

*Response:* Following the reviewer's suggestion, we have added the reference standards in the revised MS (see Line 217)

11. Line 202, "...... a small portion of...."? How much?

*Response:* We noted the value of the portion (8%) in the revised MS (see Line 239).

12. Lines 289-290, So the wood combustion is not belonging to biomass burning?

*Response:* Yes, the wood combustion also includes under biomass burning. We modified it in the revised MS (see Lines 321-322).

13. Lines 291, 322 "......several times......" "......several times abundant..." How much?

*Response:* We replaced the word "several" with the value in the revised MS (see Lines 323 and 355)

14. Line 408, "......the $NO_3^-$ is more susceptible for decomposition at higher temperatures......" so the $NO_3^-$ decomposed to what? Which process?

*Response:* At high temperatures, $NH_4NO_3$ decomposed into gaseous $HNO_3$ and $NH_3$ (Russell et al., 1983). We noted this point in the revised MS (see Lines 442-443).

**References:**

Please see the citations in the List of References in the revised MS.

---

## Referee Report (RR1)

Title: Measurement Report: Chemical components and 13C and 15N isotope ratios of fine aerosols over Tianjin, North China: Year-round observations
Author(s): Zhichao Dong et al.
MS No.: acp-2022-291
MS type: Measurement report

**General comments:**

This is a quite sufficient measurement report, the amount of data from different parameters is quite large. Therefore, it is easy to mask the main idea of the story. The authors tried to reveal the source and atmospheric processes of fine aerosols in Tianjin region based on the measurements of chemical components and stable isotopes of carbon and nitrogen. However, in my opinion, it is still hard to easily capture the main idea of the story when reading through the whole MS in current version. After the revision, the MS has indeed been improved. However, I still think the writing and structure of the MS require further improvement to make it much easier and clearer for readers to understand. Detailed comments could be found as follows:

**Specific Comments:**

For the whole structure of the "Results and Discussion", I think it might be better to show as the following orders?

(1)Meteorology and backward air mass trajectories; (2) Concentration and seasonal variations of $PM_{2.5}$; (3) Concentration and seasonal variations of carbonaceous components; (4) Implications for $PM_{2.5}$ sources through relationships and mass ratios of carbonaceous components. In this section, I do think the relationship between $PM_{2.5}$ concentrations and carbonaceous components should be added, because for example, the authors have explained that the "EC directly emits from incomplete combustion of fossil fuels and biomass burning", therefore, the relationships between $PM_{2.5}$ concentrations and EC should be a clear indicator for the source of $PM_{2.5}$, but such kind of relationship is not shown in current version, so as the relationship with other carbonaceous components. And, I'm quite confusing with the relationship between WIOC and SOC in current version, why only the relationship between WIOC and SOC was shown? (5) Implications for $PM_{2.5}$ sources through $\delta^{13}C_{TC.}$ In this section, would it be better to summarize the $\delta^{13}C$ of different sources (Fig. 11) into several types? There are too many different sources in current version, it is hard to compare; (6) Concentration and seasonal variations of nitrogenous components and other inorganic ions. In current version of MS, the authors introduced $NH_4^+$ and $NO_3^-$ concentrations in section 3.4, while introduced water-soluble nitrogenous components in section of 3.5. This is confusing because the $N-NH_4^+$ and $N-NO_3^-$ also belong to water-soluble nitrogen. (7) Implications for $PM_{2.5}$ sources through relationships of nitrogen components and other inorganic ions. (8) Implications for $PM_{2.5}$ sources through $\delta^{15}N_{TN}$. Summarize the $\delta^{15}N$ of different sources (Fig. 12) into several types?

The section of "Ionic balance" in current version is better to delete, cause I did not see any importance of this section on revealing the source and atmospheric processes of $PM_{2.5}$ in current description.

**Technical corrections:**

Lines 36-53: The authors introduced the EC, OC, SOC and WSOC in order, and then introduce EC and OC again, it is kind of circle, why don't put the two sections of EC and OC together?

Lines 85-88: Better show the range of the sources.

Lines 88-90: Better explain how the isotopic fractionation affects the isotope values of carbon and nitrogen.

Line 92: Add reference after "……are significant".

Lines 92-101: Kind of confusing, better make it clear, especially how to use $\delta^{13}C$ and $\delta^{15}N$ to investigate the aging process.

Lines 107-108: I don't understand why there are two different area of forest in Tianjin (2039 and 1364)? Better shown in percentage instead.

Line 120: Add "$SO_4^{2-}$, $Ca^{2+}$, $Mg^{2+}$……" after "inorganic ions"

Lines 147-148: I didn't buy it, cause the temperature could be more than ~30°C in summer, this will still have minor effects on the samples?

Line 184: Explain which ions?

Table 1: No units.

Lines 287-289: Seems this sentence belong to section 3.2.

Lines 438-439: Seems belong to section 3.4.3.

Line 466: Should be Fig. 8.

Lines 498-499: Why don't put the $\delta^{13}C_{TC}$ of fatty acids into Figure 11?

---

## Referee Report (RR2)

**Title: Measurement Report: Chemical components and 13C and 15N isotope ratios of fine aerosols over Tianjin, North China: Year-round observations**
Author(s): Zhichao Dong et al.
MS No.: acp-2022-291
MS type: Measurement report

**General comments:**
After revision, this manuscript has been improved a lot. However, honestly, there are some sections still needs to be further clarified. Detailed comments could be found as follows:

**Specific Comments:**
Generally, I'm confused by two points that proposed by the authors.

The first one is still, about the application of isotopes to trace the source contributions. In introduction section, the authors stated that the isotopic fractionation is more significant in the case of the isotopic composition of molecular species, but insignificant in the case of $\delta^{13}C_{TC}$ and $\delta^{15}N_{TN}$. However, the authors then added a sentence "unless gas-to-particle and/or particle-to-gas transitions are significant". So, to my understanding, the authors actually indicated that the fractionation effects could be significant on the values of $\delta^{13}C_{TC}$ and $\delta^{15}N_{TN}$ in aerosols. If so, then the authors can not directly use the values of $\delta^{13}C_{TC}$ and $\delta^{15}N_{TN}$ in aerosols to explore the source contributions in Sections 3.5 and 3.8, unless the fractionation effect on $\delta^{13}C_{TC}$ and $\delta^{15}N_{TN}$ in Tianjin aerosols was clearly discussed.

The second one is the use of WSOC/OC in Section 3.4. I'm quite confused with the explanation of the authors. The authors explained that "…WSOC is mainly generated by oxidation reactions of VOCs in the atmosphere, rather than primary emissions", and stated that "…the mass fraction of WSOC in OC can be regarded as an indicator of aging of aerosols in the atmosphere". Then, the authors also added a sentence that "…when the contribution of the WSOC is insignificant from biomass burning". However, according to the former paragraph and whole manuscript, the biomass burning is a key contributor to the aerosol components in Tianjin aerosols. So the WSOC/OC can still be used to explore the aging of aerosols? Why? The authors need to clarify this.

**Technical corrections:**
Line 22: I don't think you have direct evidence to support the idea "…they were mainly driven by $NO_3$ radicals in the former period"

Line 26: Add $SO_4^{2-}$, $NO_3^-$ and $NH_4^+$ following sulfate, nitrate and ammonium, respectively. And double check the whole manuscript to define such abbreviations when they are first referred.

Line 35: Probably better to add a sentence to make it more meaningful, such as "Therefore, it is important to explore the source and formation process of the $PM_{2.5}$".

Line 36: change ":" to ",". And, double check whether the "-" (through the whole manuscript) is in English style.

Line 40: Replace "(secondary OC, SOC)" with "to form secondary OC (SOC)"?

Line 41: Add "," following "37%".

Lines 71-74: I think better move these sentences to lines 127.

Lines 123-125: I think the authors should focus on why trace the source and formation process in Tianjin aerosol is important, but not what they are stating in current version.

Line 160: The authors better color the area of Tianjin instead of using a red star.

Line 218: Explain how the 0.83 was obtained.

Line 223: IRMS

Line 234: why 5 days??

Line 236: The authors should add some sentences about the descriptions of statistical analysis and remember to adjust the title of the 2.3 Section correspondingly.

Lines 263-264: Add citations.

Line 289: Replace "fruitful" with "effective".

Line 312: Delete "and".

Line 339: "40,0"??

Lines 375-378: Why the high concentrations of $NO_3^-$ in aerosols could accelerate the oxidation reaction of VOCs by $NO_3$ radicals? Which process? Clarify it.

Figure 5. Add p-value after each regression function.

Line 402: Wrong number of -6.5.

Line 404: Double check whether the "±" is in English style.

Lines 455-458: Why the increased emission in winter is not regarded as the reason for the peaked $NO_3^-$ concentration in winter?

Lines 461-462: Why the oceanic source of $SO_4^{2-}$ is not regarded as the reason for higher concentrations of $SO_4^{2-}$ in summer than spring?

Lines 493-494: Keep consistent of "p", and double check the style of "≥".

Lines 572-574: I don't think the authors can draw such conclusion based on the discuss above, cause there is no discussions on the seasonal change of $\delta^{15}N_{TN}$ at all.

---

## Author Response (AR2)

We thank the editor and referees' for their critical reading of the manuscript and constructive comments and suggestions, which helped to improve the quality of the MS. The MS is revised according to all the comments.

**Editor**

Please address the comments from the reviewer, especially for the writing and structure of the MS.

*Response:* We revised the manuscript according to all the comments and suggestions of the referee #2, mainly the writing and structure of the MS. Please see our point-by-point responses below and the revisions in the revised MS.

**Referee #2**

**General comments:**

This is a quite sufficient measurement report, the amount of data from different parameters is quite large. Therefore, it is easy to mask the main idea of the story. The authors tried to reveal the source and atmospheric processes of fine aerosols in Tianjin region based on the measurements of chemical components and stable isotopes of carbon and nitrogen. However, in my opinion, it is still hard to easily capture the main idea of the story when reading through the whole MS in current version. After the revision, the MS has indeed been improved. However, I still think the writing and structure of the MS require further improvement to make it much easier and clearer for readers to understand. Detailed comments could be found as follows:

*Response:* We thank the reviewer once again for his/her appreciation of our work and comments/suggestions. The MS is revised according to all the comments from the referee, and the point-by-point responses are provided below.

**Specific Comments:**

For the whole structure of the "Results and Discussion", I think it might be better to show as the following orders?

(1)Meteorology and backward air mass trajectories; (2) Concentration and seasonal variations of $PM_{2.5}$; (3) Concentration and seasonal variations of carbonaceous components; (4) Implications for $PM_{2.5}$ sources through relationships and mass ratios of carbonaceous components. In this section, I do think the relationship between $PM_{2.5}$ concentrations and carbonaceous components should be added, because for example, the authors have explained that the "EC directly emits from incomplete combustion of fossil fuels and biomass burning", therefore, the relationships between $PM_{2.5}$ concentrations and EC should be a clear indicator for the source of $PM_{2.5}$, but such kind of relationship is not shown in current version, so as the relationship with other carbonaceous components. And, I'm quite confusing with the relationship between WIOC and SOC in current version, why only the relationship between WIOC and SOC was shown? (5) Implications for $PM_{2.5}$ sources through $\delta^{13}C_{TC}$. In this section, would it be better to summarize the $\delta^{13}C$ of different sources (Fig. 11) into several types? There are too many different sources in current version, it is hard to compare; (6) Concentration and seasonal variations of nitrogenous components and other inorganic ions. In current version of MS, the authors introduced $NH_4^+$ and $NO_3^-$ concentrations in section 3.4, while introduced watersoluble nitrogenous components in section of 3.5. This is confusing because the $N-NH_4^+$ and $N-NO_3^-$ also belong to water-soluble nitrogen. (7) Implications for $PM_{2.5}$ sources through relationships of nitrogen components and other inorganic ions. (8) Implications for $PM_{2.5}$ sources through $\delta^{15}N_{TN}$. Summarize the $\delta^{15}N$ of different sources (Fig. 12) into several types?

*Response:* We agree with the reviewer's opinion fully, and re-structured the 'Results and Discussion' section by dividing it into 8 sub-sections, as suggested. Please see the subsections 3.1-3.8 of 'Results and Discussion' section in the revised MS.

Following the reviewer's suggestion, we included the relationships between $PM_{2.5}$ and OC, EC and WSOC in order to assess the sources of $PM_{2.5}$. Please see Fig. 6D-F and Lines 330-334 and 354-355 in the revised MS.

Generally, it has been recognized that WIOC might be produced by incomplete combustion of fossil fuels and cooking activities and composed of long chain aliphatic hydrocarbons, ketones, alkanes and polycyclic aromatic hydrocarbons. However, in recent times, it has been suggested that the WIOC could also be produced by secondary processes in the atmosphere. Interestingly, we found high correlation between WIOC and SOC in autumn, winter, suggesting that the secondary formation the WIOC is significant in the Tianjin atmosphere. In order to show such findings, we confined to present the linear relations between SOC and WIOC only, rather than with WSOC, which is known to be mostly produced by secondary processes, as well. We noted these points in the revised MS (see Lines 385-394).

In fact, we summarized the $\delta^{13}C$ and $\delta^{15}N$ of various source types such as marine and continental including biomass burning and fossil fuel combustion in Introduction Section in the revised MS (see Lines 90-95, respectively). Here, we confined to compare our results with the isotopic signatures of point sources and other literature values., in order to identify the potential specific sources.

Both the inorganic ions including $NH_4^+$ and $NO_3^-$ and nitrogenous components are combined into subsection 3.6 in the revised MS. Therefore, such confusion doesn't arise now.

The section of "Ionic balance" in current version is better to delete, because I did not see any importance of this section on revealing the source and atmospheric processes of $PM_{2.5}$ in current description.

*Response:* Following the reviewer's suggestion, we removed the 'Ionic balance' section in the revised MS.

**Technical corrections:**

Lines 36-53: The authors introduced the EC, OC, SOC and WSOC in order, and then introduce EC and OC again, it is kind of circle, why don't put the two sections of EC and OC together?

*Response:* Following the reviewer's suggestion, we restructured this paragraph by first describing different sources of EC and OC followed by their atmospheric loadings and impacts, and then introducing the sources and impacts of SOC and WSOC. See Lines 36-54 in the revised MS.

Lines 85-88: Better show the range of the sources.

*Response:* Following the reviewer's suggestions, we included the range or average values of the sources in the revised MS (see Lines 90-95).

Lines 88-90: Better explain how the isotopic fractionation affects the isotope values of carbon and nitrogen.

*Response:* Following the reviewer's suggestions, we described how the isotopic fractionation occurs during the occurrence of chemical reactions and phase transitions and thus influence the corresponding isotope

ratios with aging in the revised MS (see Lines 96-101).

Line 92: Add reference after "……are significant".

*Response:* Following the reviewer's suggestions, we cited the appropriate references in the revised MS (see Line 103).

Lines 92-101: Kind of confusing, better make it clear, especially how to use $\delta^{13}C$ and $\delta^{15}N$ to investigate the aging process.

*Response:* We made it clear by adding a phrase "---, which could accelerate the enrichment of 13C and 15N in the particles, ---" in that statement in the revised MS (see Lines 106).

Lines 107-108: I don't understand why there are two different area of forest in Tianjin (2039 and 1364)? Better shown in percentage instead.

*Response:* In fact, the 2,039 km$^2$ area is the total forest land area and the other areas meant for developed and natural forest areas. However, in order avoid any confusion to the reader, we provided only the total forest area, including its % in the total Tianjin land area in the revised MS (see Lines 117-120).

Line 120: Add "SO42-, Ca2+, Mg2+……" after "inorganic ions"

*Response:* We added the list of inorganic ions measured ($Cl^-$, $SO_4^{2-}$, $NO_3^-$, $Na^+$, $NH_4^+$, $K^+$, $Mg^{2+}$ and $Ca^{2+}$) in the revised MS (see Line 131).

Lines 147-148: I didn't buy it, cause the temperature could be more than ~30°C in summer, this will still have minor effects on the samples?

*Response:* We agree with the reviewer's opinion that there will be minor effects of sampling artifacts at ~.30°C in summer. In order to address such discrepancy, we toned down our statement by adding another phrase: "----, although we do not rule out them completely", in the revised MS (see Lines 156-159).

Line 184: Explain which ions?

*Response:* We added the list of inorganic ions: $Cl^-$, $SO_4^{2-}$, $NO_3^-$, $Na^+$, $NH_4^+$, $K^+$, $Mg^{2+}$ and $Ca^{2+}$, in the revised MS (see Line 194).

Table 1: No units.

*Response:* We added the units of all parameters in both Table title and in the Table in the revised MS (see Table 1).

Lines 287-289: Seems this sentence belong to section 3.2.

*Response:* Following the reviewer's suggestions, we moved this sentence into the sub-section 3.2 in the revised MS (see Lines 262-265).

Lines 438-439: Seems belong to section 3.4.3.

*Response:* Such discrepancy doesn't arise now, because this section (3.4.3) is combined with the previous section and they appearing under the section 3.7 in the revised MS.

Line 466: Should be Fig. 8.

*Response:* We regret for the typo, and corrected it in the revised MS (see Line 520).

Lines 498-499: Why don't put the δ13CTC of fatty acids into Figure 11?

*Response:* Because this study is focused on $\delta^{13}C$ of TC and all the data provided in Fig. 9 is of only $\delta^{13}C$ of TC, we preferred to provide the $\delta^{13}C$ of fatty acids in the text rather than in the Fig. 9.

---

## Author Response (AR3)

**Author Responses to Comments**

**Notification to the authors**:

1. Please ensure that the colour schemes used in your maps and charts allow readers with colour vision deficiencies to correctly interpret your findings. Please check your figures using the Coblis – Color Blindness Simulator (https://www.color-blindness.com/coblis-color-blindness-simulator/) and revise the colour schemes accordingly.

*Response:* According to the advice, we checked Figures using the Coblis – Color Blindness Simulator and revised the color schemes.

2. Please note that the Figure #12 is unreadable. Take care of a clear image before publishing.

*Response:* We changed the Figure 12 in the revised MS in order to make its image clear.

**Referee#2 Comments**

We thank the reviewer for his/her appreciation of our work and comments/suggestions. The MS is revised according to all the comments from the referee, and the point-by-point responses are provided below.

Specific Comments:

Generally, I'm confused by two points that proposed by the authors.

The first one is still, about the application of isotopes to trace the source contributions. In introduction section, the authors stated that the isotopic fractionation is more significant in the case of the isotopic composition of molecular species, but insignificant in the case of $\delta^{13}C_{TC}$ and $\delta^{15}N_{TN}$. However, the authors then added a sentence "unless gas-to-particle and/or particle-to-gas transitions are significant". So, to my understanding, the authors actually indicated that the fractionation effects could be significant on the values of $\delta^{13}C_{TC}$ and $\delta^{15}N_{TN}$ in aerosols. If so, then the authors can not directly use the values of $\delta^{13}C_{TC}$ and $\delta^{15}N_{TN}$ in aerosols to explore the source contributions in Sections 3.5 and 3.8, unless the fractionation effect on $\delta^{13}C_{TC}$ and $\delta^{15}N_{TN}$ in Tianjin aerosols was clearly discussed.

*Response:* We thank the referee for raising this point. In fact, "the isotopic fractionation is more significant in the case of the isotopic composition of molecular species, but insignificant in the case of $\delta^{13}C_{TC}$ and $\delta^{15}N_{TN}$, because the TC and TN contents contain both the reactants and products in the particle phase and the gas-to-particle and/or particle-to-gas transitions are not intensive, except under extreme temperatures, even in the case of $NH_4^+ \leftrightarrow NH_3$". In order to avoid such confusion to the reader, we modified our statement in the introduction section, as stated above, in the revised MS (please see Lines 97-101). Therefore, the use of $\delta^{13}C_{TC}$ and $\delta^{15}N_{TN}$ in aerosols to explore the source contributions in Sections 3.5 and 3.8, remain logical and valid and do not require any further discussion about the fractionation effect on $\delta^{13}C_{TC}$ and $\delta^{15}N_{TN}$ in Tianjin aerosols.

The second one is the use of WSOC/OC in Section 3.4. I'm quite confused with the explanation of the authors. The authors explained that "…WSOC is mainly generated by oxidation reactions of VOCs in the atmosphere, rather than primary emissions", and stated that "…the mass fraction of WSOC in OC can be regarded as an indicator of aging of aerosols in the atmosphere". Then, the authors also added a sentence that "…when the contribution of the WSOC is insignificant from biomass burning". However, according to the former paragraph and whole manuscript, the biomass burning is a key contributor to the aerosol components in Tianjin aerosols. So the WSOC/OC can still be used to explore the aging of aerosols? Why? The authors need to clarify this.

*Response:* Both secondary formation from VOCs and emission from biomass burning are the two major sources of WSOC. However, the WSOC/OC has been considered as an indicator for aging of aerosols in the atmosphere when the contribution of the WSOC is relatively low or insignificant from biomass burning emission. Therefore, based on the differences in the WSOC/OC between season and season(s), and their comparability with the literature and the linear relationship between WSOC and OC, we found that the secondary formation of WSOC is more intensive in spring/summer, but its contribution from biomass burning emissions is also significant, particularly in winter and autumn. In order to make it more clear to the reader, we modified and/or added some phrases in 4th paragraph: "WSOC --- Tianjin region", clarifying these points in the revised MS (see Lines 356-376).

**Technical corrections:**

Line 22: I don't think you have direct evidence to support the idea "…they were mainly driven by NO3 radicals in the former period"

*Response:* We agree with the referee. In order to avoid any potential misleading, we tone downed it by modifying the phrase: "were mostly" to "might be", in the revised MS (see Line 22).

Line 26: Add $SO_4^{2-}$, $NO_3^-$ and $NH_4^+$ following sulfate, nitrate and ammonium, respectively. And double check the whole manuscript to define such abbreviations when they are first referred.

*Response:* Following the reviewer's suggestions, we added the $SO_4^{2-}$, $NO_3^-$ and $NH_4^+$ following sulfate, nitrate and ammonium, respectively, in the revised MS (see Line 26). Also, checked the whole manuscript and given the text for abbreviations, when they referred for first time.

Line 35: Probably better to add a sentence to make it more meaningful, such as "Therefore, it is important to explore the source and formation process of the $PM_{2.5}$".

*Response:* We added "Therefore, it is important to explore the source and formation process of the $PM_{2.5}$." in the revised MS (see Line 35).

Line 36: change ":" to ",". And, double check whether the "-" (through the whole manuscript) is in English style.

*Response:* We changed the ":" to "," in the revised MS (see Line 37).

Line 40: Replace "(secondary OC, SOC)" with "to form secondary OC (SOC)"?

*Response:* We changed the "(secondary OC, SOC)" to "to form secondary OC (SOC)", in the revised MS (see Line 41).

Line 41: Add "," following "37%".

*Response:* We added the "," following "37%", in the revised MS (see Line 42).

Lines 71-74: I think better move these sentences to lines 127.

*Response:* Following the reviewer's suggestions, we moved these sentences to Lines 122 -128 in the revised MS.

Lines 123-125: I think the authors should focus on why trace the source and formation process in Tianjin aerosol is important, but not what they are stating in current version.

*Response:* Following the reviewer's advice, we completely changed this statement as follows: "Therefore, the investigation of the Tianjin aerosols sources and formation processes provide better insights on the types of aerosol sources at regional level, in addition to the local industrial and domestic pollutant emissions in North China.", in the revised MS (see Lines 120-122), to highlight the importance of this study.

Line 160: The authors better color the area of Tianjin instead of using a red star.

*Response:* We changed the map of Tianjin in the revised MS (see Figure 1).

Line 218: Explain how the 0.83 was obtained.

*Response:* 0.83 is the propagating error in WSON estimation.

If Q is some combination of sums and differences, i.e.

$$Q = a + b + c + \cdots + c - (x + y + \cdots + z),$$
then
$$\delta Q = \sqrt{(\delta a)^2 + (\delta b)^2 + \cdots + (\delta c)^2 + (\delta x)^2 + (\delta y)^2 + \cdots + (\delta z)^2}$$

$\delta Q$ is the propagating error. (Reference a summary of error propagation.)

WSON = WSTN − IN, IN = (14/42) × $NO_3^-$ + (14/18) × $NH_4^+$, Thus,

$\delta$WSTN is 0.82, $\delta NO_3^-$ is 0.10, $\delta NH_4^+$ is 0.15, are the random error generated during the experimental operation, according to the parallel experiment calculation.

$$\delta IN = \sqrt{(\frac{14}{62})^2 \times (\delta NO_3^-)^2 + (\frac{14}{18})^2 \times (\delta NH_4^+)^2} = \sqrt{(\frac{14}{62})^2 \times (0.10)^2 + (\frac{14}{18})^2 \times (0.15)^2} = 0.12$$
$$\delta WSON = \sqrt{(\delta WSTN)^2 - (\delta IN)^2} = \sqrt{(0.82)^2 + (0.12)^2} = 0.83$$

Line 223: IRMS

*Response:* We modified "IrMS" to "IRMS" in the revised MS (see Line 224 and Line 227).

Line 234: why 5 days??

*Response:* Since Tianjin weather is influenced by East Asian monsoon, the air masses arrived in Tianjin are mostly originate at regional scale. That is why, in order to understand the potential source regions at regional level during the campaign, we selected the backward air mass trajectories for 5-day period. We noted this point in the revised MS (see Lines 236-237).

Line 236: The authors should add some sentences about the descriptions of statistical analysis and remember to adjust the title of the 2.3 Section correspondingly.

*Response:* Following the reviewer's suggestions, we briefly described the data statistical analysis and added a phrase in the subsection title accordingly in the revised MS (see Lines 233 & 237-240).

Lines 263-264: Add citations.

*Response:* We added citations in the revised MS (see the Line 269).

Line 289: Replace "fruitful" with "effective".

*Response:* We changed "fruitful" to "effective" in the revised MS (see Line 294).

Line 312: Delete "and".

*Response:* We deleted "and" in the revised MS (see Line 317).

Line 339: "40,0"??

*Response:* We changed "40,0" to "40.0" in the revised MS (see Line 344).

Lines 375-378: Why the high concentrations of $NO_3^-$ in aerosols could accelerate the oxidation reaction of VOCs by $NO_3$ radicals? Which process? Clarify it.

*Response:* Recently, based on model simulations and CIMS measurements, it has been found that NO3 radical oxidation of VOCs, particularly monoterpenes, through unimolecular reactions is one of the significant sources of SOA (Draper et al., ACS Earth & Space Chem., 3, 8, 1460-1470, 2019). We added this point with citation in the revised MS (see Line 384).

Figure 5. Add p-value after each regression function.

*Response:* We added the p-value at the Fig. 5 in the revised MS.

Line 402: Wrong number of -6.5.

*Response:* We corrected the typo: "−6.5" to "−26.5", in the revised MS (see Line 408).

Line 404: Double check whether the "±" is in English style.

*Response:* We checked whole text in the revised MS to make sure the "±" is in English style.

Lines 455-458: Why the increased emission in winter is not regarded as the reason for the peaked $NO_3^-$

concentration in winter?

*Response:* Yes, we agree with the reviewer's view. The enhanced emissions in winter could have also been made the higher levels of $NO_3^-$ than that in other seasons. In addition, due to the large temperature difference between winter and summer, the atmospheric chemical process caused by temperature should have also been made certain contribution. We added the emission contribution in the revised MS (see Lines 461-462).

Lines 461-462: Why the oceanic source of $SO_4^{2-}$ is not regarded as the reason for higher concentrations of $SO_4^{2-}$ in summer than spring?

*Response:* Yes, we agree with the reviewer's view. The oceanic source of $SO_4^{2-}$ is one of the reasons for higher concentration of $SO_4^{2-}$ in summer, and we mentioned it in the revised MS (see Lines 469-470).

Lines 493-494: Keep consistent of "p", and double check the style of "≥".
*Response:* We checked whole text in the revised MS to make sure that the "p" and "≥" are in English style and uniform.

Lines 572-574: I don't think the authors can draw such conclusion based on the discuss above, because there is no discussions on the seasonal change of $\delta^{15}N_{TN}$ at all.
*Response:* We discussed the seasonal change of $\delta^{15}N_{TN}$ in Lines 540-543 showed in Figure 7 the revised MS. The $\delta^{15}N_{TN}$ showed higher value in summer which might be caused by marine air mass and biogenic emissions, while the low value of $\delta^{15}N_{TN}$ in winter, driven by primary emission. Such seasonal changes in $\delta^{15}N_{TN}$ suggest that the aerosol N might be substantially influenced by season-specific source(s) and/or the chemical aging of N species. Therefore, we believe that the conclusion drawn here is logical. However, we modified/added the phrases to tone down the conclusion in the revised MS (see Lines 579-581).

---

## Author Response (AR4)

**Author Responses to Comments**

**Notification to the authors**:

1. With the next revision, please add the numeric code to indicate the affiliation of the authors next to the names of the authors on the title page of the *.pdf manuscript file.

*Response:* According to the advice, we added the numeric code: [1], to indicate the affiliation of the authors in the revised MS.

2. Please ensure that the colour schemes used in your maps and charts allow readers with colour vision deficiencies to correctly interpret your findings. Please check your figures using the Coblis – Color Blindness Simulator (https://www.color-blindness.com/coblis-color-blindness-simulator/) and revise the colour schemes accordingly.

*Response:* According to the advice, we checked all the Figures using the Coblis – Color Blindness Simulator and modified the color schemes of the Figs. 4-6, 8 and 10-12 accordingly in the revised MS.

**Referee#2 Comments**

We thank the reviewer for his/her appreciation of the revisions, and comments/suggestions. The MS is revised according to all the comments from the referee, and the point-by-point responses are provided below.

*General comments:*

This version of the manuscript has been further improved. However, some sentences are still bad in expressions, better find someone more professional to make it easier for readers.

*Response:* We thoroughly checked the whole text and improved the language in the revised MS.

*Specific Comments:*

1. Sentences in the manuscript are often long and hard to read, for example, lines 82-86, lines 97-101, lines 101-105, lines 274-278, lines 288-291, etc.

   *Response:* We modified/rephrased all the noted sentences in the revised MS. Please see Lines: 82-87, 97-105, 275-280 and 290-298.

2. Grammar and expression problem: lines 14-15, lines 64-66, lines 74-76, lines 127-128, lines 294-295, etc.

   *Response:* We corrected all the language errors. See Lines: 14-15, 65-67, 75-77,128-129 and 296-297, in the revised MS.

3. Line 17. It is not completely right to say "whereas SO42− was higher in summer at both the sites", cause the SO42− is still peaked in winter in ND.

   *Response:* We corrected this phrase as "--- whereas $SO_4^{2-}$ was higher in summer than in all other seasons at HEP and comparable among seasons, although it peaked in winter at ND." in the revised MS (see Lines: 16-18).

4. Lines 383-385. I can understand that the NO3 radical oxidation is an important source of SOA, however, I don't think the higher concentration of ion NO3- will accelerate the oxidation

process. If the authors do not agree, better explain the mechanism of how ion NO3- accelerate the oxidation reaction of VOCs.

*Response:* We fully agree with the reviewer that the VOCs are oxidized by $NO_3$ radical, not by $NO_3^-$ ion. In the previous version, we highlighted the loadings of $NO_3^-$ ion to indirectly represent the high abundance of $NO_2$ and thus $NO_3$ radical availability. However, in order to make it clear to the reader, we modified it by noting the seasonal variations of $NO_2$, instead of $NO_3^-$ ion, in the revised MS (see Lines: 382-387).

5. Lines 923-924, the style of the references is weird.

*Response:* We corrected the references style in the revised MS.